# MedITok: A Unified Tokenizer for Medical Image Synthesis and Interpretation

## Abstract

Advanced autoregressive models have reshaped multimodal AI. However, their transformative potential in medical imaging remains largely untapped due to the absence of a *unified* visual tokenizer—one capable of capturing fine-grained visual structures for faithful image reconstruction and realistic image synthesis, as well as rich semantics for accurate diagnosis and image interpretation. To this end, we present MedITok, the first unified tokenizer tailored for medical images, encoding both low-level structural details and high-level clinical semantics within a unified latent space. To balance these competing objectives, we introduce a novel two-stage training framework: a visual representation alignment stage that cold-starts the tokenizer reconstruction learning with a visual semantic constraint, followed by a textual semantic representation alignment stage that infuses detailed clinical semantics into the latent space. Trained on the meticulously collected large-scale dataset with over 30 million medical images and 2 million image-caption pairs, MedITok achieves state-of-the-art performance on more than 30 datasets across 9 imaging modalities and 4 different tasks. By providing a unified token space for autoregressive modeling, MedITok supports a wide range of tasks in clinical diagnostics and generative healthcare applications. Model and code are available in the supplementary material.

## 1 Introduction

The rapid evolution of advanced autoregressive (AR) models, such as GPT-4o (OpenAI, 2025), has revolutionized multimodal learning. These models excel at generating and understanding text, image, and audio data via unified processing of token-based representations. In medical imaging, AR models begin to demonstrate similar promise, powering report generation (Tanno et al., 2025), tumor segmentation (Chen et al., 2025a), counterfactual synthesis (Ma et al., 2025a), and diagnostic visual question answering (VQA) (Li et al., 2023). By translating complex biomedical image patterns into token sequences, these models can synthesize realistic images and interpret clinical cues (*e.g.*, ground-glass opacities on chest computed tomography, microcalcifications on mammography) in the images, with the potential to streamline workflows and improve patient outcomes.

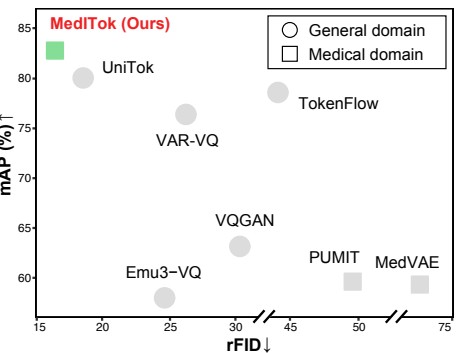

Figure 1: Performance comparison of different tokenizers on medical image reconstruction (rFID) and classification (mAP). MedITok achieves the best of both worlds.

A critical ingredient in building a powerful AR model is the *visual tokenizer*, which translates an input image to a sequence of discrete tokens suitable for AR modeling. Existing approaches can be divided into two categories. (1) Generation-oriented tokenizers optimized for pixel-level reconstruction[1], *e.g.*, VQGAN (Esser et al., 2021). These tokenizers precisely capture low-level structure in the image that is vital to image compression (Varma et al., 2025; Wang et al., 2024c) and

---

[1]In this paper, "reconstruction" refers to autoencoding reconstruction: decoding an input image from its latent representation.

generation (Zhu et al., 2024; Sun et al., 2024; Yu et al., 2024; Yao et al., 2025). However, they do not explicitly encode discriminative features and are therefore not suitable for interpreting the concepts and objects embedded in the image. (2) Interpretation-driven tokenizers trained with discriminative objectives, *e.g.*, CLIP (Radford et al., 2021). These tokenizers excel at capturing high-level textual semantics, making them indispensable for visual comprehension, but they fail to accurately retain spatial structures and textures in the image.

**Motivation.** Visual tokens that embed only one side of this structure-semantic spectrum will offload the representation learning burden onto downstream AR models, which often incurs heavy pre-training costs and can still leave either side under-utilized (Wang et al., 2024b; Chen et al., 2025b). These limitations are especially acute in the medical domain, where clinical tasks typically demand both precise visual structures and clinical semantics. However, current medical image tokenizers tend to specialize in one single aspect (Luo et al., 2023b; Zhang et al., 2023b), which lacks a unified, information-rich token space and thereby limits the potential of downstream medical AR models for accurate, interpretable, and data-efficient diagnosis.

Our goal is to democratize a foundation visual tokenizer for medical images. Nonetheless, training a unified tokenizer for medical images poses unique challenges: (1) A naïve joint optimization of visual reconstruction and textual semantic objectives often causes mutual interference and degraded performance (Wu et al., 2025; Qu et al., 2024). (2) Paired image-caption data for training is much more scarce in the medical domain, compared to the abundant unlabeled images.

To addresses these issues, we propose a novel two-stage training framework. Instead of directly coupling the visual reconstruction and textual semantic, it involves a *visual representation alignment stage* to first establish basic semantic awareness with strong reconstruction capabilities as a cold-start, followed by the *textual semantic alignment stage* for learning finer-grained semantic information. This framework leads to our model: MedITok, the first unified visual tokenizer tailored for medical images. MedITok encodes both low-level structural information, supporting image synthesis and compression, and high-level semantics, enabling medical image interpretation and multimodal comprehension, serving as a general foundation for diverse community use.

Specifically, the first training stage cold-starts MedITok on pure medical images, optimizing for reconstruction fidelity with a light semantic constraint on the latent space. Then, the textual semantic alignment stage tunes MedITok on image-caption pairs, enhancing semantic richness by aligning visual tokens to textual embeddings of detailed captions. This approach allows MedITok to *effectively encode structural and semantic information* while *exploiting both unpaired medical images and image-text pairs at scale*, making a unified token space to develop powerful AR models for diverse tasks. To achieve this, we meticulously collect a large-scale dataset comprising over 30 million medical images and 2 million image-caption pairs from more than 300 public sources, with broad coverage of imaging modalities, anatomies, and pathologies. This collection ensures that MedITok learns robust representations for medical image synthesis and interpretation.

**Contributions.** **(1)** We propose a novel training framework for developing a unified visual tokenizer, which effectively scales up with medical image and text data and progressively builds a unified token space. **(2)** We introduce MedITok, the first medical image tokenizer that unifies the encoding of structural details and clinical semantics. **(3)** Extensive experimental results on over 30 datasets, spanning 9 imaging modalities, across 4 different tasks, showcase MedITok's state-of-the-art performance over existing approaches and broad applicability to diverse medical tasks. **(4)** Model and code will be open-source. Data access links are provided respecting all original licenses.

## 2 RELATED WORK

**AR Models in Medical Vision Tasks.** AR models have shown impressive scalability and generalizability in general vision-language processing. In medical domain, these models have been extended to a variety of tasks: image captioning and VQA for interpreting scans and assist diagnosis (Li et al., 2023; Moor et al., 2023; Chen et al., 2024c), lesion segmentation model across different imaging modalities (Chen et al., 2025a), medical image synthesis for counterfactual analysis (Ma et al., 2025a) and modality transfer (Ren et al., 2024), *etc.* More recently, HealthGPT (Lin et al., 2025) further unifies both medical image synthesis and comprehension capabilities within an AR framework for broader applications. However, these methods typically general-domain tokenizers pre-trained on

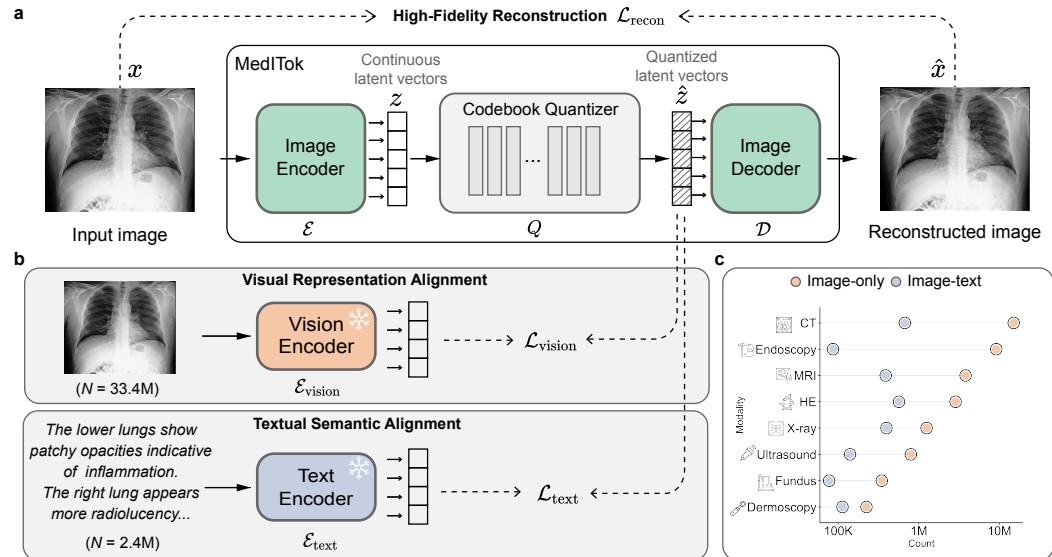

Figure 2: Overview of the proposed training framework. (a) Architecture of MedITok. (b) Two-stage training: visual representation alignment with pretrained visual semantics, followed by textual semantic alignment using clinical captions. (c) Statistics across modalities for our training data.

natural images, which encode insufficient clinical knowledge and capture either low-level structural detail or high-level clinical concepts, rarely both, whereas clinical tasks usually demand joint representation. To this end, we introduce MedITok, the first unified tokenizer tailored for medical images to support a wide range of tasks and empower advanced AR models in the medical field.

**Unified Visual Tokenizers.** Visual tokenizers encode images into token sequences suitable for AR modeling. Recent works (Wu et al., 2025; Ma et al., 2025b; Qu et al., 2024) seek to unify the encoding of both low-level details and high-level semantics into one single visual tokenizer, enhancing the multimodal generation and comprehension capabilities of downstream AR models. TokenFlow (Qu et al., 2024) proposes an intuitive dual-codebook design that explicitly decouples semantic and pixel-level cues. UniTok (Ma et al., 2025b) instead shows that simply scaling codebook capacity lets a single token space capture both, offering a more scalable solution. In medical imaging, recent research such as MedVAE (Varma et al., 2025) builds high-fidelity continuous latent compressors but stops short of providing unified tokens for downstream AR modeling. Our approach is the first medical-domain tokenizer to fill this gap, supplying unified token space to power the next generation of medical multimodal models.

## 3 METHODOLOGY

By encoding both low-level structural details and high-level clinical semantics, MedITok directly supports medical image reconstruction and classification tasks, and can be further integrated into AR models for more complex tasks, *e.g.*, medical image synthesis and interpretation, *etc.* Below, we start with a preliminary on the image tokenization (Sec. 3.1) and provide detailed description of our model and training framework (Sec. 3.2) and dataset curation process (Sec. 3.3).

### 3.1 PRELIMINARY

The drive to apply powerful autoregressive models from natural language processing to visual data has spurred the development of image tokenization techniques, converting images into sequences of visual tokens. Among these, Vector Quantization (VQ)-based approaches (Van Den Oord et al., 2017; Esser et al., 2021) are foundational.

In a typical VQ-based image tokenizer, an image $x$ is first mapped by an encoder $\mathcal{E}$ to a spatial grid of latent vectors $z \in \mathbb{R}^{h \times w \times d}$. Each vector in this grid is then quantized by assigning it to the closest entry in a learned, finite codebook $\mathcal{C} = \{c_k\}_{k=1}^{K}$, where $c_k \in \mathbb{R}^d$ represents a visual token and $K$

is the codebook size. The quantized grid of latent vectors, $z_q \in \mathbb{R}^{h \times w \times d}$, effectively represent the image as a compressed sequence of visual tokens. A decoder $\mathcal{D}$ is then trained to reconstruct the image from these representations, producing $\hat{x} = \mathcal{D}(z_q)$. During training, the encoder $\mathcal{E}$, decoder $\mathcal{D}$, and the codebook $\mathcal{C}$ are jointly optimized. It typically involves a composite loss function designed to ensure both accurate reconstruction and effective codebook learning (Esser et al., 2021), defined as:

$$\mathcal{L}_{\mathrm{recon}}(\hat{x}, x, z_q, z) = \mathcal{L}_{\mathrm{image}}(\hat{x}, x) + \lambda_{\mathrm{comm}}\mathcal{L}_{\mathrm{comm}}(z_q, z), \qquad (1)$$

where $\mathcal{L}_{\mathrm{image}}$ is the image fidelity loss consisting of a mean square error loss, a perceptual loss (Johnson et al., 2016), and an adversarial loss, encouraging high-fidelity reconstructions. The commitment loss (Van Den Oord et al., 2017) $\mathcal{L}_{\mathrm{comm}}$ ensures the encoder outputs $z$ to commit to their nearest codebook vectors. Our work builds upon these foundational principles of VQ-based tokenization but introduces a novel training framework tailored to unified medical image tokenization.

## 3.2 MEDITOK TRAINING FRAMEWORK

A unified visual tokenizer must reconcile two objectives that naturally compete: preserving low-level spatial detail for image reconstruction and synthesis, and learning a high-level semantic token space for image interpretation. Previous works (Wu et al., 2025; Ma et al., 2025b) combine visual reconstruction and textual representation learning objectives in one go. Such training can lead to representation collapse or suboptimal trade-offs (Qu et al., 2024). Moreover, they typically rely on large-scale image-caption pairs while overlooking the abundance of unpaired images. We propose a novel two-stage training framework to train our unified visual tokenizer MedITok, unlocking the potential of unlabeled images in the medical domain and progressively transitioning from reconstruction-focused learning to unified token learning.

As depicted in Fig. 2, MedITok is comprised of an image encoder $\mathcal{E}$, a quantizer $Q$, and a decoder $\mathcal{D}$. Our framework begins with a *visual representation alignment* stage, which cold-starts the training of the image encoder $\mathcal{E}$ and a decoder $\mathcal{D}$ using a vast corpus of unpaired medical images. The primary focus is on capturing low-level structural information, guided by only a light semantic constraint from a pretrained vision encoder $\mathcal{E}_{\mathrm{vision}}$. Subsequently, in the second stage termed *textual semantic alignment*, $\mathcal{E}$ is refined using high-quality image-caption pairs. Here, the emphasis shifts towards enhancing the semantic richness of the learned tokens by aligning them with clinical captions processed by a pretrained text encoder $\mathcal{E}_{\mathrm{text}}$. This approach not only alleviates the conflicts between reconstruction and semantic learning objectives but also allows us to effectively leverage large-scale unpaired images alongside paired image-text data for unified tokenizer training.

**Visual Representation Alignment.** Given an input image $x$, the encoder $\mathcal{E}$ produces continuous latent vectors $z$, which are then quantized by the quantizer $Q$ to yield discrete latent vectors $z_q = Q(z)$. The decoder $\mathcal{D}$ then learns to reconstruct the image $\hat{x} = \mathcal{D}(z_q)$. The pretrained vision encoder $\mathcal{E}_{\mathrm{vision}}$ encodes the image $x$ into a semantic representation, which is then projected into the space of $z_q$ via a linear layer $f_{\mathrm{vision}}$ to provide semantic supervision for learning $z_q$. We use a composite loss function for training, defined as:

$$\mathcal{L}_{\mathrm{stage1}} = \mathcal{L}_{\mathrm{recon}}(\hat{x}, x, z_q, z) + \lambda_{\mathrm{vision}}\mathcal{L}_{\mathrm{vision}}(z_q, f_{\mathrm{vision}}(\mathcal{E}_{\mathrm{vision}}(x))), \qquad (2)$$

where $\mathcal{L}_{\mathrm{vision}}$ is a contrastive loss that imposes light semantic constraint on the latent space, with the factor $\lambda_{\mathrm{vision}}$ set to 0.1. By prioritizing reconstruction while gently guiding the latent space with pre-trained visual semantics, this stage ensures MedITok develops a robust understanding of visual structure, preparing it for fine-grained semantic alignment in the subsequent stage.

**Textual Semantic Alignment.** This stage focuses on enhancing the semantic richness of the learned image tokens and aligning them with fine-grained textual representations extracted from detailed medical captions. The training in this stage is driven by the following loss function:

$$\mathcal{L}_{\mathrm{stage2}} = \mathcal{L}_{\mathrm{recon}}(\hat{x}, x, z_q, z) + \lambda_{\mathrm{text}}\mathcal{L}_{\mathrm{text}}(z_q, f_{\mathrm{text}}(\mathcal{E}_{\mathrm{text}}(t))), \qquad (3)$$

where $t$ denotes the detailed caption of the image $x$, and $f_{\mathrm{text}}$ is another linear layer. $\mathcal{L}_{\mathrm{text}}$ is the contrastive loss, with a balancing factor $\lambda_{\mathrm{text}}$ set to 1. This stage further integrates the structural and semantic representation learning, empowering MedITok for a wide range of downstream medical applications requiring nuanced understanding.

### 3.3 DATASET CURATION

The development of MedITok necessitates a comprehensive and diverse dataset. To this end, we undertake an extensive data collection effort, aggregating medical images and image-text pairs from over 300 publicly available sources. For example, image-text pairs are collected from BIOMED-ICA (Lozano et al., 2025), MedICaT (Subramanian et al., 2020), MIMIC-CXR (Johnson et al., 2019), ROCOv2 (Rückert et al., 2024), PMC-OA (Lin et al., 2023), MM-Retinal (Wu et al., 2024), and GMAI-MM-Caption-1.7M (Li et al., 2024) datasets.

Quality control is a critical step in our data collection pipeline to ensure that the training data are of sufficient quality for learning meaningful representations. We employ a combination of automated and manual filtering to exclude images of low quality or limited clinical relevance. Specifically, an image is excluded if, after proxy RGB conversion, it meets any of the following criteria: (1) low pixel intensity range below 50; (2) insufficient resolution, where the smallest dimension is under 128 pixels; (3) low information content, indicated by a standard deviation of pixel values below 10; (4) limited palette, with three or fewer unique pixel values; (5) unrelated content, such as tables, plots, or non-clinical images extracted from publications. For text data, we only retain captions pertinent to clinical imaging, determined by the tags within each dataset or clinical keyword matching.

These checks efficiently remove noisy and uninformative samples and ensures higher quality input for our training framework, resulting in a massive corpus of 33,428,922 medical images for the visual representation alignment stage, and 2,422,827 high-quality medical image-caption pairs for the textual semantic alignment stage. This collection encompasses over eight imaging modalities, including computed tomography (CT), dermoscopy, endoscopy, fundus photography, magnetic resonance imaging (MRI), pathology, ultrasound, and X-ray, spanning a wide spectrum of anatomical regions and pathological findings. We leave detailed sources and statistics in our Appendix A.

## 4 EXPERIMENTS

In this section, we present comprehensive experiments to evaluate the proposed MedITok across four different task families, including medical image reconstruction, medical image classification, modality-conditioned medical image synthesis, and medical visual question answering. Since each task is evaluated using specialized metrics appropriate to its goals, we introduce them within each corresponding subsection.

### 4.1 EXPERIMENTAL SETUP

**Datasets.** (1) For medical image reconstruction, we collect images from 23 publicly available datasets (McCollough et al., 2017; Landman et al., 2015; Heimann et al., 2009; Kawahara et al., 2018; Giotis et al., 2015; Ali et al., 2022; Kiranyaz et al., 2020; Cartucho et al., 2024; Ali et al., 2020; Decencière et al., 2014; Ovreiu et al., 2021; Fraz et al., 2012; Hoover et al., 2000; Graham et al., 2019b; Da et al., 2022; Nir et al., 2018b; Bao et al., 2025; Pati et al., 2020; Pedraza et al., 2015; Al-Dhabyani et al., 2020; Lian et al., 2021; Halabi et al., 2019; Tabik et al., 2020), totaling 35,736 images covering 8 imaging modalities. (2) For medical image classification, we benchmark on five subsets of the latest MedMNIST collection (Yang et al., 2023) in different imaging modalities, including BreastMNIST (Al-Dhabyani et al., 2020) for ultrasound, DermaMNIST (Tschandl et al., 2018; Codella et al., 2019) for dermoscopy, PathMNIST (Kather et al., 2019) for pathology images, PneumoniaMNIST (Kermany et al., 2018) for chest X-ray, and RetinaMNIST (Liu et al., 2022) for fundus photography, where all images are resized to 256×256. (3) For modality-conditioned medical image synthesis, we employ data from BloodMNIST (Acevedo et al., 2020), BreastMNIST, ChestMNIST (Wang et al., 2017b), DermaMNIST, PathMNIST, and RetinaMNIST to train and test the downstream AR image synthesis models. (4) Finally, for medical visual question answering, we use PubMedVision (Chen et al., 2024c) dataset to train the downstream multimodal language models, and evaluate them on two widely used medical visual question answering (VQA) benchmarks: VQA-RAD (Lau et al., 2018) and SLAKE (Liu et al., 2021a). We carefully conduct manual cross-checking on the data used for evaluating and training MedITok, minimizing the risk of data leakage. Please see Appendix B for more details on statistics and tasks.

**Implementation Detail.** We build MedITok with a hybrid ViT architecture (Chen et al., 2024b) using PyTorch (Paszke et al., 2019), and implement the quantizer with 8 codebooks, each containing

Table 1: Medical image reconstruction across different imaging modalities using different models. The best results are highlighted in **bold** and the second best results are underlined. SSIM values are presented as percentages. $f_d$ denotes the downsampling factor. "↓": The lower the better.

| Metrics | Models | $f_d$ | CT | Dermo. | Endo. | Fundus. | MRI | Patho. | US | X-ray | Avg. | Avg. rank |
|---|---|---|---|---|---|---|---|---|---|---|---|---|
| rFID (↓) | VQGAN | 8 | 15.97 | 33.57 | 27.33 | 27.22 | 21.33 | 67.68 | 29.48 | 18.66 | 30.16 | 4.9 |
| | Emu3-VQ | 8 | 11.83 | 27.91 | 20.83 | 16.27 | 13.52 | 69.89 | 25.43 | 11.99 | 24.71 | 3.4 |
| | VAR-VQ | 16 | 14.69 | 30.27 | 19.74 | 21.69 | 13.99 | 70.06 | 26.09 | 12.18 | 26.09 | 4.1 |
| | TokenFlow | 16 | 24.78 | 44.28 | 47.42 | 34.93 | 26.81 | 98.22 | 51.77 | 24.51 | 44.09 | 7.0 |
| | UniTok | 16 | 9.27 | 23.15 | 13.64 | 16.22 | 9.30 | 47.77 | 20.93 | 8.61 | 18.61 | 2.0 |
| | PUMIT | 16 | 32.67 | 53.46 | 56.22 | 27.51 | 25.43 | 142.98 | 37.04 | 23.78 | 49.88 | 7.1 |
| | MedVAE | 8 | 20.17 | 140.39 | 114.00 | 117.39 | 23.34 | 123.20 | 30.60 | 11.54 | 73.64 | 6.5 |
| | MedITok | 16 | **7.88** | **22.27** | **10.66** | **14.39** | **6.32** | **46.54** | **17.64** | **6.55** | **16.53** | **1.0** |
| PSNR (↑) | VQGAN | 8 | 31.13 | 29.28 | 25.60 | 35.40 | 29.54 | 20.42 | 24.79 | 31.68 | 28.48 | 6.3 |
| | Emu3-VQ | 8 | 36.11 | 31.68 | 28.96 | **39.64** | 34.32 | 22.08 | 27.57 | 35.81 | **32.02** | 2.6 |
| | VAR-VQ | 16 | 31.32 | 29.26 | 25.75 | 35.73 | 29.83 | 20.86 | 25.22 | 31.10 | 28.63 | 5.8 |
| | TokenFlow | 16 | 28.64 | 27.23 | 23.72 | 33.45 | 27.68 | 19.33 | 23.26 | 28.71 | 26.50 | 7.8 |
| | UniTok | 16 | 33.60 | 30.97 | 27.55 | 37.21 | 31.50 | 22.18 | 26.97 | 32.97 | 30.34 | 4.3 |
| | PUMIT | 16 | 33.64 | 30.23 | 29.08 | 37.33 | 33.13 | 23.09 | 28.31 | 33.89 | 31.09 | 3.1 |
| | MedVAE | 8 | **36.46** | 20.67 | 25.04 | 15.31 | **34.42** | 19.58 | 28.29 | **36.23** | 27.01 | 4.5 |
| | MedITok | 16 | 36.32 | **31.69** | **29.19** | 37.72 | 33.55 | **23.54** | **28.49** | 34.42 | 31.74 | 1.8 |
| SSIM (↑) | VQGAN | 8 | 88.51 | 75.28 | 76.84 | 92.32 | 84.39 | 48.42 | 68.18 | 91.14 | 78.14 | 6.8 |
| | Emu3-VQ | 8 | 92.79 | 79.34 | 84.71 | 94.33 | 95.72 | 54.70 | 75.14 | **95.54** | 83.78 | 3.5 |
| | VAR-VQ | 16 | 89.51 | 76.69 | 79.21 | 93.08 | 93.68 | 47.40 | 69.99 | 90.79 | 80.04 | 6.0 |
| | TokenFlow | 16 | 82.43 | 67.19 | 69.47 | 89.60 | 90.22 | 33.09 | 56.56 | 84.50 | 71.63 | 7.8 |
| | UniTok | 16 | 92.42 | 81.00 | 84.47 | 94.45 | 95.47 | 56.42 | 76.40 | 92.74 | 84.17 | 3.9 |
| | PUMIT | 16 | 92.10 | 85.41 | 87.81 | 94.60 | 96.59 | 63.81 | 81.46 | 94.52 | 87.04 | 2.6 |
| | MedVAE | 8 | 92.86 | 75.32 | 81.52 | 69.46 | 95.92 | 53.10 | 77.45 | 94.77 | 80.10 | 4.4 |
| | MedITok | 16 | **93.73** | **85.47** | **88.99** | **95.27** | **97.22** | **65.99** | **83.93** | 95.39 | **88.25** | 1.1 |

4,096 eight-dimensional latent vectors. We train MedITok using AdamW (Loshchilov & Hutter, 2019) optimizer for 3 epochs in the first stage and 2 epochs in the second stage, with a global batch size of 512. Image resolution is of $256 \times 256$. The encoder of MedITok is initialize with weights from UniTok for efficient training. We choose BiomedCLIP (Zhang et al., 2023b) as the pretrained semantic vision and text encoders for alignment in our training framework, which is frozen throughout the training. Detailed setup can be found in our Appendix C.

**Competing Tokenizers.** We compare MedITok with powerful visual tokenizers from both general and medical domains, including VQGAN (Esser et al., 2021), Emu3-VQ (Wang et al., 2024b), VAR-VQ (Tian et al., 2024), TokenFlow (Qu et al., 2024), UniTok (Ma et al., 2025b), PUMIT (Luo et al., 2023b), and MedVAE (Varma et al., 2025). VQGAN, Emu3-VQ, and VAR-VQ are pure VQ-based tokenizers trained on natural images without semantic alignments, yet showing great promise in building medical multimodal models (Lin et al., 2025; Ma et al., 2025a). TokenFlow and UniTok are two state-of-the-art unified image tokenizers in the general domains. PUMIT and MedVAE are two medical visual tokenizers that mainly focus on fine-grained detail preservation.

## 4.2 MEDICAL IMAGE RECONSTRUCTION

We employ reconstruction Fréchet inception distance (rFID) (Heusel et al., 2017), peak signal-to-noise ratio (PSNR), and structural similarity index measure (SSIM) (Wang et al., 2004) to evaluate the image reconstruction performance. Notably, Woodland et al. (2024) have shown that ImageNet-pretrained feature extractors are more consistent and aligned with human medical expert judgment than their counterparts pretrained on medical images, and we follow their work to implement rFID.

Quantitative results are shown in Table 1. MedVAE struggles on the modalities with colored imaging (*e.g.*, fundus photography) as it is trained only on grayscale images (Varma et al., 2025). Notably, despite with a large downsampling factor of $16\times$, MedITok delivers the best overall reconstruction quality across 8 modalities, outperforming other tokenizers including those with only $8\times$ downsampling. This highlights the efficiency of MedITok in balancing compression with reconstruction fidelity. Fig. 3 visualizes images reconstructed by different tokenizers and corresponding error maps. MedVAE fails to preserve colors due to limited generalizability, while UniTok discards nuanced details. By contrast, our MedITok consistently preserves fine-grained structures and color fidelity. Please refer to Appendix D for more results.

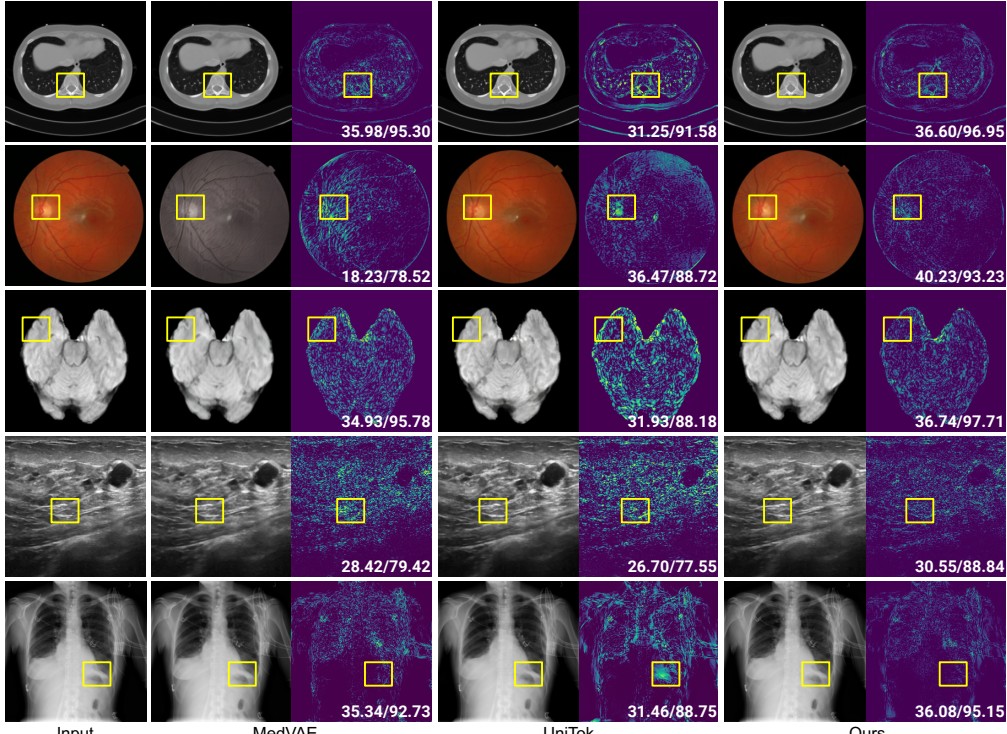

Figure 3: Reconstruction results across multiple imaging modalities. Each reconstructed image is paired with an absolute error map against the input image with PSNR/SSIM values.

Table 2: Downstream image classification performance (mAP / AUC) with linear probing setup. The best results are highlighted in **bold** and the second best results are underlined. Values are presented as percentages.

| Models | Dermoscopy | Fundus | Pathology | Ultrasound | X-ray | AVG |
|---|---|---|---|---|---|---|
| VQGAN | 35.71/85.97 | 41.59/77.33 | 72.69/94.57 | 73.29/76.35 | 91.34/93.32 | 62.92/85.51 |
| Emu3-VQ | 30.79/82.88 | 38.90/71.71 | 42.57/82.75 | 82.65/85.30 | 92.75/93.29 | 57.53/83.19 |
| VAR-VQ | 58.76/94.02 | 51.71/85.53 | 90.80/98.31 | 87.31/89.06 | 97.56/97.79 | 77.23/92.94 |
| TokenFlow | 61.78/93.50 | 52.07/83.77 | 95.21/99.23 | **88.19**/88.12 | 97.69/98.03 | 78.99/92.53 |
| UniTok | 66.16/94.60 | 55.94/85.05 | 96.63/99.49 | 87.34/88.60 | 95.98/96.84 | 80.41/92.92 |
| PUMIT | 23.64/71.92 | 36.60/72.87 | 81.52/96.50 | 68.81/73.67 | 88.80/91.64 | 59.87/81.31 |
| MedVAE | 37.66/85.26 | 39.31/75.29 | 48.02/84.85 | 77.74/82.36 | 95.41/95.47 | 59.54/84.64 |
| MedITok (ours) | **71.52/95.60** | **56.41/86.88** | **96.88/99.60** | 87.45/**89.07** | **99.08/99.19** | **82.27/94.07** |

## 4.3 MEDICAL IMAGE CLASSIFICATION

To assess whether a visual tokenizer encodes clinical semantics in the latent space, we adopt a linear-probing (Alain & Bengio, 2016) protocol on a suite of medical image classification tasks (Yang et al., 2023). Specifically, we freeze each tokenizer and append a linear layer to its encoder, training the linear layer to convergence on the image classification task and report the performance in terms of mean average precision (mAP) and area under the ROC curve (AUC) on the corresponding test sets. Results are presented in Table 2. Models optimized purely for image reconstruction (*e.g.*, Emu3-VQ, PUMIT) degrade on tasks where fine-grained clinical semantics are required for nuanced classification, *e.g.*, retinal disease grading in fundus photographs. General-domain unified tokenizers like TokenFlow and UniTok show improved but limited performance. By contrast, our MedITok encodes rich clinical-relevant semantics and delivers the best overall classification performance, showing that rich semantic information is embedded in MedITok's unified token space.

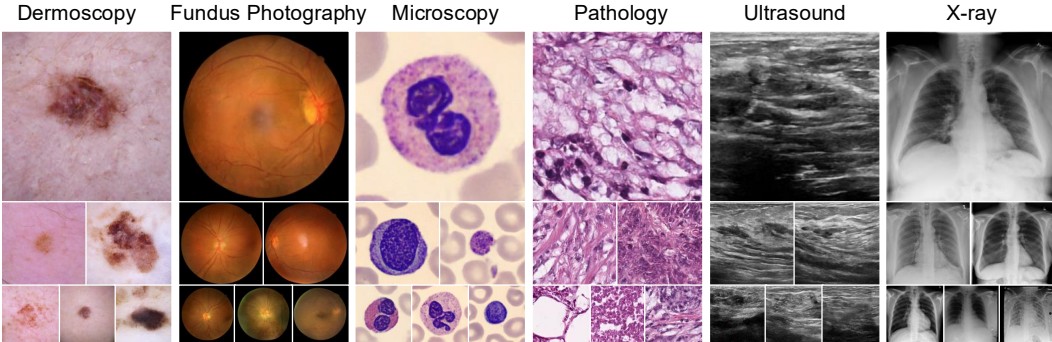

| Dermoscopy | Fundus Photography | Microscopy | Pathology | Ultrasound | X-ray |

Figure 4: Modality-conditioned synthesized image examples produced by our LlamaGen$_{\text{MedITok}}$.

Table 3: Modality-conditioned medical image generation performance.

| Models | gFID ($\downarrow$) | Diversity ($\uparrow$) |
|---|---|---|
| LlamaGen$_{\text{VQGAN}}$ | $130.93_{\pm 3.58}$ | $0.6503_{\pm 0.03}$ |
| LlamaGen$_{\text{UniTok}}$ | $80.71_{\pm 3.18}$ | $0.6584_{\pm 0.02}$ |
| LlamaGen$_{\text{MedITok-S1}}$ | $94.98_{\pm 1.89}$ | $0.6479_{\pm 0.02}$ |
| LlamaGen$_{\text{MedITok}}$ | $\mathbf{76.78}_{\pm 1.91}$ | $\mathbf{0.6883}_{\pm 0.01}$ |

Table 4: Visual question answering accuracy on two medical benchmarks.

| Models | VQARAD | SLAKE-val | SLAKE-test |
|---|---|---|---|
| LLaVA-Med | $43.90_{\pm 2.88}$ | $40.30_{\pm 2.28}$ | $38.73_{\pm 3.53}$ |
| LLaVA-Med$_{\text{UniTok}}$ | $49.66_{\pm 1.11}$ | $44.44_{\pm 2.04}$ | $43.84_{\pm 1.28}$ |
| LLaVA-Med$_{\text{MedITok-S1}}$ | $46.56_{\pm 1.67}$ | $40.73_{\pm 1.52}$ | $41.02_{\pm 0.83}$ |
| LLaVA-Med$_{\text{MedITok}}$ | $\mathbf{52.99}_{\pm 2.14}$ | $\mathbf{49.02}_{\pm 3.45}$ | $\mathbf{48.09}_{\pm 1.42}$ |

## 4.4 MEDICAL IMAGE SYNTHESIS

We explore applying unified visual tokenizers to image synthesis task by incorporating each tokenizer in the LlamaGen (Sun et al., 2024) framework for modality-conditioned medical image synthesis, including six imaging modalities: dermoscopy, fundus photography, microscopy, pathology images, ultrasound, and X-ray. Specifically, we build two LlamaGen models using MedITok-S1, a variant of MedITok that only goes through the first training stage, and MedITok. These two models, denoted by "LlamaGen$_{\text{MedITok-S1}}$" and "LlamaGen$_{\text{MedITok}}$", respectively, are compared with other LlamaGen variants with different visual tokenizers, *i.e.*, "LlamaGen$_{\text{VQGAN}}$" and "LlamaGen$_{\text{UniTok}}$". We follow previous work (Bluethgen et al., 2024) to report generation Fréchet inception distance (gFID) and diversity score for evaluating the fidelity and the diversity of the synthesized images. For visual diversity metric, we first sample $N$ images from the modality-to-image model for each modality, and calculate $\sum_{i \neq j}(1 - \text{MS-SSIM}(\boldsymbol{x}_i, \boldsymbol{x}_j))/\binom{2}{N}$ for all distinct pairs $(\boldsymbol{x}_i, \boldsymbol{x}_j)$ among $N$ synthesized images, where MS-SSIM denotes the multi-scale structural similarity index (Wang et al., 2003). The overall diversity score is defined as the mean diversity score over all imaging modalities.

Quantitative results in Table 3 show that LlamaGen using general-domain tokenizer like VQ-GAN or UniTok struggles with high-quality medical image generation. Notably, LlamaGen$_{\text{MedITok}}$ achieves the best visual fidelity and diversity. We also note that LlamaGen$_{\text{MedITok}}$ surpasses LlamaGen$_{\text{MedITok-S1}}$ by a non-trivial margin, indicating that textual semantic alignment may regularize the token space and boost the image synthesis task. Fig. 4 illustrates images synthesized by LlamaGen$_{\text{MedITok}}$ across various modalities, presenting realistic structures and textures of biological tissues and organs. Note that, although MedITok is not trained on microscopy modalities, it still supports realistic synthesis of microscopy images. Please refer to Appendix D for more examples.

## 4.5 MEDICAL IMAGE INTERPRETATION

We further evaluate the effectiveness of different tokenizers in medical image interpretation by integrating each as the image encoder in the LLaVA-Med (Li et al., 2023) framework, yielding three models: LLaVA-Med$_{\text{UniTok}}$, LLaVA-Med$_{\text{MedITok-S1}}$, and LLaVA-Med$_{\text{MedITok}}$. We initialize the language backbone using the released weights of LLaVA-Med, train these models on the PubMed-Vision (Chen et al., 2024c) dataset, and evaluate their accuracy on two widely used medical VQA benchmarks: VQA-RAD (Lau et al., 2018) and SLAKE (Liu et al., 2021a).

Table 5: Ablation studies of MedITok. "#Img": number of images used in the first training stage, "#Img-txt": number of image-text pairs used in the second training stage.

| Idx. | Vision Target Repr. | Text Target Repr. | Objective | #Img | #Img-txt | PSNR | SSIM | mAP | AUC |
|------|---------------------|-------------------|-----------|------|----------|------|------|-----|-----|
| (i)   | CLIP-V       | –             | Contrast | 800k  | –    | 30.99 | 86.67 | 70.80 | 89.01 |
| (ii)  | BiomedCLIP-V | –             | Contrast | 800k  | –    | 30.00 | 83.85 | 78.35 | 92.23 |
| (iii) | BiomedCLIP-V | BiomedCLIP-T  | Contrast | 800k  | 1M   | 30.03 | 84.32 | 80.09 | 92.64 |
| (iv)  | BiomedCLIP-V | –             | Contrast | 1.8M  | –    | 31.38 | 84.36 | 78.49 | 92.25 |
| (v)   | BiomedCLIP-V | BiomedCLIP-T  | Contrast | 800k  | 2.4M | 29.74 | 84.14 | 80.28 | 92.72 |
| (vi)  | BiomedCLIP-V | BiomedCLIP-T  | Contrast | 2M    | 2.4M | 30.20 | 85.50 | 82.23 | 93.61 |
| (vii) | BiomedCLIP-V | BiomedCLIP-T  | Contrast | 33.4M | 2.4M | 31.74 | 88.25 | 82.27 | 94.07 |

As shown in Table 4, LLaVA-Med equipped with our final MedITok consistently outperforms other variants across all benchmarks. We observe a similar improvement from MedITok-S1 to MedITok as in Table 2, indicating the necessity of the textual semantic alignment stage. The underperformance of general-domain tokenizer, UniTok, highlights the importance of domain-specific semantic encoding. These results confirm that MedITok provides effective representations to develop powerful AR models for downstream medical image interpretation tasks.

### 4.6 ABLATION STUDIES

**Choice of Pre-trained Encoder.** Ideally, the pretrained encoders in the proposed training framework are designed to be flexible, provided they offer rich semantic representations, *e.g.*, CLIP-family (Radford et al., 2021; Zhang et al., 2023b). We experiment with two options: the general-domain CLIP (Radford et al., 2021) and the biomedical-domain BiomedCLIP (Zhang et al., 2023b). Results in Rows (i) and (ii) of Table 5 show that, by aligning to the representations produced from BiomedCLIP, MedITok achieves significant improvement in the medical image classification tasks while maintaining competitive image reconstruction performance, indicating that domain-specific pre-trained encoders can provide clinically-relevant semantics that benefit downstream medical tasks.

**Two-Stage Training.** We further validate the importance of the textual semantic alignment stage by comparing our full two-stage framework, shown in Row (iii) of Table 5, against a single-stage variant with the same number of training images, shown in Row (iv). Our two-stage approach boosts image classification without degrading reconstruction quality, highlighting that the textual representation alignment stage helps the model capture richer cross-modal semantics.

**Image Data Scaling.** One notable benefit of our proposed training framework is that it allows effective use of unpaired medical image datasets, which are typically more accessible than image-text data. Rows (v), (vi), and (vii) of Table 5 illustrate the impact of scaling up the number of unpaired image corpus in the first training stage. Notably, expanding the image data from 800k to 33.4M yields consistent improvements across all metrics, demonstrating the scalability of our proposed approach, allowing it to fully exploit medical image data to enhance both structural fidelity and downstream diagnostic performance. More experimental results can be found in Appendix D.

## 5 CONCLUSION

In this paper, we propose MedITok, a unified medical image tokenizer that encodes both low-level structural details and high-level clinical semantics. Leveraging a novel two-stage training framework which involves visual representation alignment on large-scale unpaired images and textual semantic alignment on high-quality image-caption pairs, MedITok learns a unified token space that facilitates medical image reconstruction, classification, synthesis, and VQA, outperforming existing general-domain and medical-domain models across multiple imaging modalities. By providing a unified token space, we believe MedITok will serve as a foundational building block for next-generation multimodal models in medical applications. Please refer to Appendix E for more discussion.

**Ethics Statement.** We affirm adherence to the ICLR Code of Ethics. This work uses only publicly available datasets with clear licensing; no new human or animal subjects were recruited and no protected health information beyond what is already de-identified in the source data was used. We discuss potential societal risks in Appendix E.4, including bias, misuse of generative models, and the need for oversight. Large language models were used only to aid and polish wording, improving the flow and clarity of the presentation; they did NOT generate analyses, experiments, figures, or results, and all technical content was authored by the authors.

**Reproducibility Statement.** We provide anonymized code and access to model weights in our supplementary material. Training data sources, preprocessing, and statistics are detailed in Appendix A with dataset lists (Tables S5–S8). Evaluation datasets, task definitions, and metrics appear in Sec. 4 and Appendix B (including Tables S10 and S11). Experimental setups are detailed in Sec. 4 and Appendix C.

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

# Appendix of Paper:
# "MedITok: A Unified Tokenizer for Medical Image Synthesis and Interpretation"

CONTENTS

Figure S1: Overview of the training data for MedITok. Left: exemplar images used in the first training stage. Right: word cloud generated from the captions used in the second training stage.

*Return to:* **Introduction** | **Experiments** | **Appendix Contents**

## A  TRAINING DATASET

In this section, we provide a comprehensive overview of the training dataset used in this work, including the collection (Appendix A.1), preprocessing (Appendix A.2), and statistics (Appendix A.3) of image-only datasets and image-text paired datasets. The construction of this training dataset is pivotal to the success of our proposed MedITok, as it ensures a diverse and high-quality representation of medical images and text descriptions across multiple modalities, anatomical regions, and clinical contexts.

### A.1  DATA COLLECTION

We begin by identifying and collecting medical imaging datasets from over 300 publicly available sources, ensuring broad coverage of imaging modalities and clinical scenarios. Our selection criteria include: **(1)** Appropriate Licensing: We only select datasets with clear licensing, ensuring compliance with data usage policies; **(2)** Clinical Relevance: Only datasets that provide diagnostic-quality images or clinically annotated images were included; and **(3)** Diversity in Imaging Modalities and Anatomies: We prioritize datasets that collectively cover a wide range of anatomical regions and pathologies.

### A.2  DATA PREPROCESSING

#### A.2.1  EXTRACTING 2D IMAGES FROM 3D VOLUMES

A significant portion of our dataset comprises volumetric medical images (CT and MRI). To fully utilize these data to train our 2D visual tokenizer, we carefully convert them into 2D image slices using a modality-specific preprocessing strategy.

**CT images extracted from volumes.**  Each 3D CT volume is first converted to Hounsfield Units (HU) using the rescaling slope and intercept recorded in the metadata, and is then clipped to the range of $[-1000, 2000]$. To obtain 2D slices from the 3D volume, we extract slices along three orthogonal planes (axial, coronal, and sagittal), and select every fifth slice along each plane. We then perform an initial quality filtering by retaining CT slices that met all the following criteria: (1) a background ratio (the proportion of pixels with HU values $\leq -1000$) $\leq 0.6$, (2) a valid body ratio (the proportion of pixels with HU values $\geq -300$) $\geq 0.1$, and (3) a pixel intensity standard deviation $< 100$. These criteria ensure the removal of largely empty slices with minimal anatomical content.

Note that, we save the CT images extracted from 3D volumes in their original HU values without scaling them to the $[0, 255]$ range. By doing so, we can apply various CT window settings on the CT images during model training as a form of data augmentation, as detailed in Appendix C.1.

**MRI images extracted from volumes.** We process each MRI volume by clipping voxel values to the $[0.5^{\text{th}}, 99.5^{\text{th}}]$ percentile range, followed by min-max normalization to $[-1, 1]$. The 2D slices are extracted using the same way as CT preprocessing. The initial quality filtering for MRI excludes those slices with mean pixel values $\leq -0.9$ or standard deviation $\leq 0.2$.

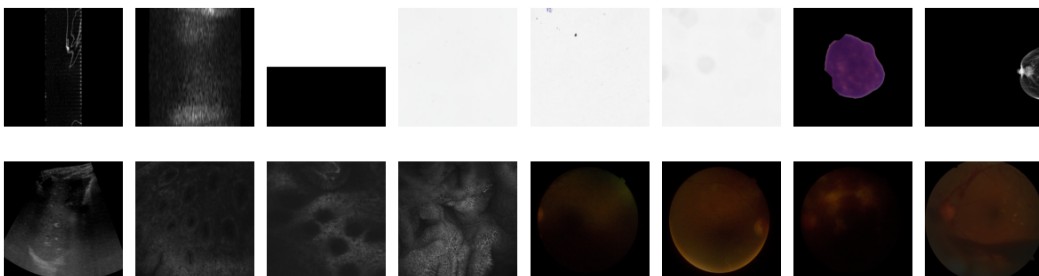

Figure S2: Low-quality images filtered by our quality control pipeline.

### A.2.2 QUALITY CONTROL

Once we obtain all the 2D images, we implement the following process to ensure that only high-quality, clinically relevant images are included in the training dataset:

- Dynamic Range Check: Images with pixel intensity ranges below 50 were filtered out to ensure adequate contrast.
- Resolution Filtering: Images with a minimum dimension below 128 pixels were excluded to maintain structural integrity.
- Information Content Validation: Images with low standard deviation (below 10) in pixel values were discarded.
- Palette Limitation Removal: Images with three or fewer unique pixel values were removed.
- Relevance Verification: Non-clinical images, such as tables, plots, or irrelevant illustrations, were manually screened and excluded.

For instance, the "Relevance Verification" is mainly applied on the BIOMEDICA (Lozano et al., 2025) dataset, which originally contains approximately 24,050,423 image-text pairs extracted from biomedical publications. Each image-text pair is tagged with primary and secondary labels. We retain only those pairs with a primary label of "Clinical Imaging" and a secondary label matching one of the following: "computerized tomography", "clinical imaging", "light microscopy", "immuno-histochemistry", "endoscopy", "eye", "X-ray radiography", "ultrasound", "magnetic resonance", "brain", "skin lesion", and "mammography". Image-text pairs tagged with irrelevant secondary labels (*e.g.*, "scientific illustration", "ambiguous", "plot", "diagram", *etc.*) are all excluded. Such filtering significantly reduces the BIOMEDICA dataset from 24,050,423 to 1,216,529 image-text pairs for use in our experiments.

Following the automated checks described above, we perform a manual quality assessment by randomly sampling 10 images from each dataset for manual visual inspection. If any low-quality outliers are identified, we further examine other images from the corresponding dataset to evaluate overall quality. Finally, we try our best to remove the images that share the same sources with data in our benchmarking datasets in Appendix B.

Fig. S2 displays some low-quality images detected by the dynamic range check, information content validation, and palette limitation removal. For another example, Fig. S3 shows images that are not tagged as "clinical imaging" in the original BIOMEDICA (Lozano et al., 2025) dataset.

### A.3 DATA STATISTICS

After the collection and the preprocessing, we present detailed sources and image counts of our "image-only" dataset, which is used in the first training stage of MedITok, in Tables S5–S8. The details of the "image-caption" dataset, used in the second training stage, are presented in Table S9.

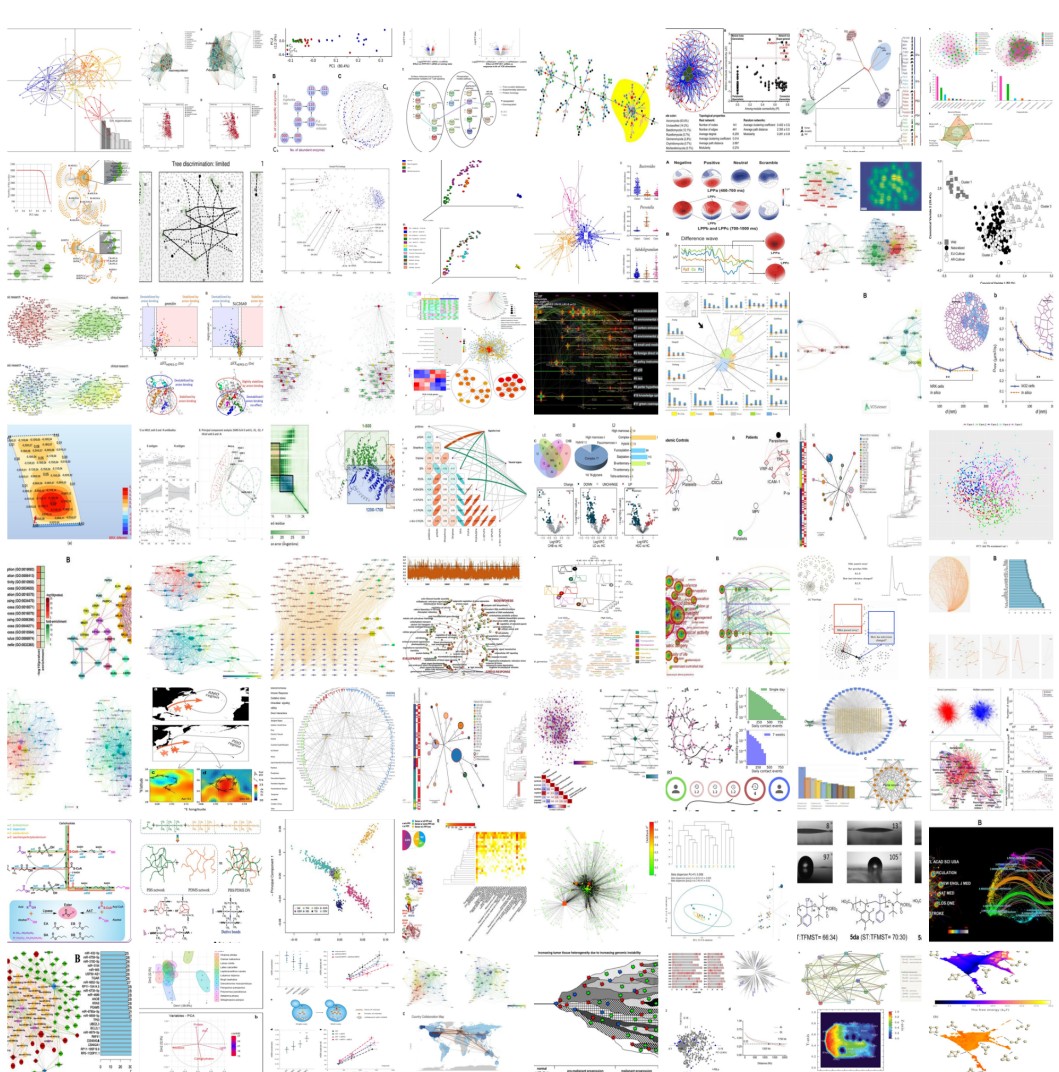

Figure S3: Irrelevant images filtered out by our quality control pipeline.

## B  BENCHMARKING DATASETS

This section outlines the datasets used for evaluating the performance of MedITok across four core tasks: medical image reconstruction (Appendix B.1), classification (Appendix B.2), modality-conditioned image synthesis (Appendix B.3), and visual question answering (Appendix B.4). We tried our best to avoid any overlap or data leakage between the training data of MedITok and these benchmark datasets.

### B.1  IMAGE RECONSTRUCTION

To assess the reconstruction capabilities of MedITok, we curated a high-quality evaluation set of 35,736 images spanning 8 imaging modalities. These images are collected from 23 publicly available datasets, as detailed in Table S10. Importantly, all images used for evaluation are strictly excluded from the training corpus to prevent any overlap. This evaluation set reflects a diverse distribution of anatomical structures, imaging protocols, and clinical contexts, enabling robust testing of image fidelity and structural preservation.

### B.2  IMAGE CLASSIFICATION

We adopt five subsets from the latest version[2] of MedMNIST (Yang et al., 2023) benchmark to evaluate the semantic encoding quality of the visual tokens produced by different tokenizers. These include:

- BreastMNIST (Al-Dhabyani et al., 2020) (ultrasound): binary classification of benign vs. malignant tumors.
- DermaMNIST (Tschandl et al., 2018; Codella et al., 2019) (dermoscopy): 7-way classification of skin lesions.
- PathMNIST (Kather et al., 2019) (pathology): 9-class colorectal cancer tissue types.
- PneumoniaMNIST (Kermany et al., 2018) (X-ray): pneumonia detection in chest radiographs.
- RetinaMNIST (Liu et al., 2022) (fundus): diabetic retinopathy grading.

The original images in each benchmark are of a resolution of $224 \times 224$, and are resized to $256 \times 256$ resolution for consistency with the tokenizer input. These tasks collectively test the extent to which the visual tokenizer encodes discriminative, clinically meaningful semantics. Detailed training and test split can be found in Table S11.

### B.3  IMAGE SYNTHESIS

To evaluate the generative capability of downstream autoregressive models built on top of MedITok, we conduct experiments on modality-conditioned image synthesis. Specifically, we use six subsets from the latest MedMNIST collection (Yang et al., 2023), including BloodMNIST (Acevedo et al., 2020) for microscopy, BreastMNIST (Al-Dhabyani et al., 2020) for ultrasound, ChestMNIST (Wang et al., 2017b) for chest x-ray, DermaMNIST (Tschandl et al., 2018; Codella et al., 2019) for dermoscopy, PathMNIST (Kather et al., 2019) for pathology images, and RetinaMNIST (Liu et al., 2022) for fundus photography. We gather the training partition of these subsets with their imaging modality labels to construct the training data for the downstream medical image synthesis models, which are trained to generate images conditioned on modality labels.

### B.4  VISUAL QUESTION ANSWERING

To test the utility of different visual tokenizers for medical image interpretation in multimodal settings, we benchmark on two widely adopted datasets for visual question answering (VQA) task:

---

[2]https://doi.org/10.5281/zenodo.10519652

(1) VQA-RAD (Lau et al., 2018): A radiology-specific VQA dataset with natural language questions and answers grounded in diagnostic images. We use its test set containing 451 question-answer pairs for evaluation. (2) SLAKE (Liu et al., 2021a): A multi-modal, bilingual medical VQA benchmark with more diverse imaging modalities and question types. The validation set (SLAKE-val) with 2,094 questions and test set (SLAKE-test) with 2,099 questions are adopted in our experiments.

To train vision-language model for medical image interpretation (*i.e.*, LLaVA-Med (Li et al., 2023) variants with different visual tokenizers as the image encoder), we use the PubMedVision (Chen et al., 2024c) dataset, which consists of high-quality image-question-answer triplets derived from medical publications. All VQA benchmarks are held out from the training set to ensure fair and unbiased evaluation.

---

*Return to:* **Introduction** | **Experiments** | **Appendix Contents**

---

## C EXPERIMENTAL SETUPS

In this section, we first describe the detailed implementation and training setup of MedITok (Appendix C.1) and its downstream applications (Appendix C.2) on four core tasks: image reconstruction, image classification, image synthesis, and visual question answering.

### C.1 IMPLEMENTATION DETAILS

**Architecture.** MedITok consists of a ViTamin-based (Chen et al., 2024b) image encoder and decoder, with a multi-codebook vector quantizer (Ma et al., 2025b) in the bottleneck. The encoder produces a 2D grid of latent representations, which are discretized using 8 parallel codebooks, each with 4,096 eight-dimensional vectors, resulting in a total vocabulary size of 32,768. The decoder reconstructs the image from quantized latent vectors.

**Training of MedITok.** Both training stages (*i.e.*, visual representation alignment, and textual semantic alignment) share the same reconstruction loss defined as follows:

$$\mathcal{L}_{\text{recon}}(\hat{\boldsymbol{x}}, \boldsymbol{x}, \boldsymbol{z}_{\text{q}}, \boldsymbol{z}) = \mathcal{L}_{\text{image}}(\hat{\boldsymbol{x}}, \boldsymbol{x}) + \lambda_{\text{comm}} \mathcal{L}_{\text{comm}}(\boldsymbol{z}_{\text{q}}, \boldsymbol{z}), \tag{S1}$$

$$\mathcal{L}_{\text{image}}(\hat{\boldsymbol{x}}, \boldsymbol{x}) = \|\hat{\boldsymbol{x}} - \boldsymbol{x}\|_2^2 + \lambda_{\text{adv}} \mathcal{L}_{\text{adv}}(\hat{\boldsymbol{x}}, \boldsymbol{x}) + \lambda_{\text{perc}} \mathcal{L}_{\text{perc}}(\hat{\boldsymbol{x}}, \boldsymbol{x}), \tag{S2}$$

$$\mathcal{L}_{\text{comm}}(\boldsymbol{z}_{\text{q}}, \boldsymbol{z}) = \|\boldsymbol{z}_{\text{q}} - \text{sg}[\boldsymbol{z}]\|_2^2 + \beta \|\text{sg}[\boldsymbol{z}_{\text{q}}] - \boldsymbol{z}\|_2^2, \tag{S3}$$

where $\mathcal{L}_{\text{adv}}$ denotes the adversarial loss (Esser et al., 2021), $\mathcal{L}_{\text{perc}}$ the perceptual loss (Johnson et al., 2016), and $\mathcal{L}_{\text{comm}}$ the commitment loss (Van Den Oord et al., 2017). "sg[·]" denotes the stop-gradient operation. We follow the default setting of VQGAN (Esser et al., 2021) to set $\lambda_{\text{adv}}$ as an adaptive weight and fix $\beta = 0.25$, $\lambda_{\text{perc}} = 1$, and $\lambda_{\text{comm}} = 1$. The discriminator involved in computing $\mathcal{L}_{\text{adv}}$ adopts the DINOv2 (Oquab et al., 2023) architecture. We use the AdamW (Loshchilov & Hutter, 2019) optimizers for both MedITok and the discriminator, with betas of $(0.9, 0.95)$ and a weight decay of $0.02$ for MedITok, and $(0.5, 0.9)$ and $0.2$ for the discriminator. The learning rate is initialized at $5 \times 10^{-4}$ and decayed to $5 \times 10^{-5}$ via cosine annealing; for the discriminator, it starts at $2 \times 10^{-5}$ and decays to $2 \times 10^{-6}$. The two-stage full-data training took approximately 4 days on 8 NVIDIA H100 GPUs.

We employ random resized cropping, random image flipping, random image rotation for data augmentation in the first training stage. For CT image input in HU values, we further introduce **CT windowing augmentation**, which randomly applies the following windows on the HU values: full window ($[-1000, 2000]$ HU, probability $p = 0.2$), common window ($[-1000, 1000]$ HU, $p = 0.3$), soft tissue window ($[-150, 250]$ HU, $p = 0.3$), lung window ($[-1400, 200]$ HU, $p = 0.15$), and bone window ($[-500, 1300]$ HU, $p = 0.05$).

### C.2 DOWNSTREAM TASKS

**Medical image classification.** For classification tasks, we evaluate the discriminative power of the learned visual tokens through a linear probing protocol (Alain & Bengio, 2016). Specifically, for a pretrained visual tokenizer (*e.g.*, MedITok), we only use its image encoder and quantizer, keep

them frozen, and append a single linear layer on top of the quantizer. Given an image, the image encoder produces the continuous feature maps, which are then discretized to a grid of visual tokens and are flattened to feed the linear layer for image classification. The linear layer is trained using the Adam (Kingma, 2014) optimizer with a learning rate of $10^{-4}$ and a batch size of 128. Since the tokens produced by different tokenizers lead to different convergence speed for the linear layer, we train each linear layer until convergence and report the peak performance for a fair comparison.

**Medical image synthesis.** For image synthesis, we integrate the visual tokenizer with LlamaGen-B (Sun et al., 2024), an autoregressive model designed for image generation, with 12 transformer layers, 12 heads, and 768 token dimension. We first tokenize each training image, producing a discrete token sequence. Then, LlamaGen is trained to autoregressively predict the token sequence conditioned on a modality label token. LlamaGen models are optimized using AdamW (Loshchilov & Hutter, 2019) with betas of $(0.9, 0.95)$, a weight decay of $0.05$, and a learning rate of $10^{-4}$. The models are trained for 200 epochs with a batch size of 128. We do not employ advanced strategy for sampling (*e.g.*, classifier-free guidance) and synthesize images with a temperature parameter of 1.

**Medical visual question answering.** For VQA, we adapt LLaVA-Med (Li et al., 2023) by replacing its image encoder with different visual tokenizers, followed by a projector to produce visual embeddings compatible with the pretrained language backbone[3]. We follow the staged training procedure of original LLaVA-Med, which includes a pretraining stage for the projector (with all other components frozen) and a fine-tuning stage for the language model using LoRA (Hu et al., 2022). The pretraining is conducted on 500k image-caption pairs in PubmedVision (Chen et al., 2024c) dataset for one epoch with batch size 32, while the fine-tuning takes two epochs on the 100k visual question-answer pairs.

*Return to:* **Introduction** | **Experiments** | **Appendix Contents**

# D ADDITIONAL RESULTS

This section compiles extended evidence to complement the main results, including additional ablation studies (Appendix D.1), adaptation to 3D medical volumes (Appendix D.2), additional visual Turing test for medical image synthesis (Appendix D.3), analyses of the differences between codebooks across training stages (Appendix D.4) and representative failure cases (Appendix D.5), comparison of data scale and inference efficiency (Appendix D.6), and additional visualizations for reconstruction, synthesis, and VQA (Appendix D.7), including qualitative generative and VQA examples that illustrate behavior beyond aggregate metrics.

## D.1 ADDITIONAL ABLATION STUDIES

We present additional ablation studies in Table S1 to further investigate the effectiveness of our data quality control and the proposed training framework.

**Separating Rather Than Combining Two Stages.** In contrast to previous works, we propose incorporating an extra training stage (*e.g.*, visual represenation alignment) in the training of unified visual tokenizer. A natural idea question comes: can we combine this stage and the subsequent textual semantic alignment stage together in one stage? That is, in each iteration, we optimize the following loss function:

$$\mathcal{L} = \mathcal{L}_{\text{recon}}(\hat{\boldsymbol{x}}, \boldsymbol{x}, \boldsymbol{z}_{\text{q}}, \boldsymbol{z})$$
$$+ \lambda_{\text{vision}} \mathcal{L}_{\text{vision}}(\boldsymbol{z}_{\text{q}}, f_{\text{vision}}(\mathcal{E}_{\text{vision}}(\boldsymbol{x}))) \tag{S4}$$
$$+ \lambda_{\text{text}} \mathcal{L}_{\text{text}}(\boldsymbol{z}_{\text{q}}, f_{\text{text}}(\mathcal{E}_{\text{text}}(\boldsymbol{t}))),$$

In Rows (i) and (ii) of Table S1, we empirically compare combined single-stage and our two-stage training under the same setting. The combined-stage training only slightly improves semantic metrics but significantly degrades reconstruction quality. This may be attributed to the dominance of semantic alignment objectives, which in turn escalates the inherent conflicts between reconstruction (low-level) and semantic (high-level) alignment objectives. In contrast, we use the visual representation learning

---

[3] https://huggingface.co/microsoft/llava-med-v1.5-mistral-7b

Table S1: More ablation studies of MedITok. "#Img": number of images used in the first training stage, "#Img-txt": number of image-text pairs used in the second training stage. "BiomedCLIP-T (combined)": textual semantic alignment is combined with the visual representation alignment as one single stage. "BiomedCLIP-T$^\dagger$": the BiomedCLIP (Zhang et al., 2023b) text encoder is activated during training.

| Idx. | Vision Target Repr. | Text Target Repr. | $\lambda_{\text{vision}}$ | #Img | #Img-txt | PSNR | SSIM | mAP | AUC |
|---|---|---|---|---|---|---|---|---|---|
| (i) | BiomedCLIP-V | BiomedCLIP-T (combined) | 0.1 | 800k | 1M | 29.20 | 83.22 | 81.10 | 91.97 |
| (ii) | BiomedCLIP-V | BiomedCLIP-T | 0.1 | 800k | 1M | 30.03 | 84.32 | 80.09 | 92.64 |
| (iii) | – | BiomedCLIP-T | 0 | 800k | 24M (all BIOMEDICA) | 32.23 | 89.36 | 57.97 | 76.98 |
| (iv) | – | BiomedCLIP-T | 0 | 800k | 1M (filtered BIOMEDICA) | 32.55 | 89.49 | 63.29 | 81.68 |
| (v) | BiomedCLIP-V | BiomedCLIP-T | 0.1 | 800k | 1M | 30.03 | 84.32 | 80.09 | 92.64 |
| (vi) | BiomedCLIP-V | BiomedCLIP-T | 1 | 800k | 1M | 29.99 | 83.02 | 82.00 | 91.81 |
| (vii) | – | BiomedCLIP-T | 0 | – | 2.4M | 29.06 | 79.61 | 80.29 | 91.25 |
| (viii) | – | BiomedCLIP-T | 0 | – | 2.4M (+800k)$^*$ | 30.05 | 82.12 | 80.06 | 91.18 |
| (ix) | BiomedCLIP-V | BiomedCLIP-T | 0.1 | 800k | 2.4M | 29.74 | 84.14 | 80.28 | 92.72 |
| (x) | BiomedCLIP-V | BiomedCLIP-T | 0.1 | 2M | 2.4M | 30.20 | 85.50 | 82.23 | 93.61 |
| (xi) | BiomedCLIP-V | BiomedCLIP-T$^\dagger$ | 0.1 | 33.4M | 2.4M | 34.03 | 91.05 | 51.41 | 69.84 |
| (xii) | BiomedCLIP-V | BiomedCLIP-T | 0.1 | 33.4M | 2.4M | 31.74 | 88.25 | 82.27 | 94.07 |
| (xiii) | BiomedCLIP-V | – | Cos. sim | 800k | – | 30.18 | 84.01 | 66.19 | 85.77 |
| (xiv) | BiomedCLIP-V | – | Contrast | 800k | – | 30.00 | 83.85 | 78.35 | 92.23 |

$^*$: we convert 800k pure images to pseudo image-text pairs by tagging each image with a short caption "This is a ${modality}$ image."

as a cold-start to transit from a reconstruction-based tokenizer to a unified tokenizer more smoothly, improving joint optimization of these competing objectives.

We also note that separating two stages provides more flexibility, particularly when training with significantly imbalanced data collections in the medical domain, where unlabeled images are far more abundant than image-text pairs (14x in our final training set). A staged design allows us to exploit such imbalanced data effectively and provides engineering flexibility for making modifications to the pretrained encoders (*e.g.*, adding trainable parameters), while avoiding potential gradient issues caused by heterogeneous batches.

**Data Quality Control.** Rows (iii) and (iv) of Table S1 presents the result from our pilot study to evaluate the effectiveness of our data quality control pipeline. We pretrain MedITok with pure reconstruction objective in the first training stage, and continue the second training stage on the BIOMEDICA (Lozano et al., 2025) dataset.

Specifically, in Row (iii), we adopt all 24M image-text pairs in this dataset, while in Row (iv), we use a much smaller subset with approximately 1M pairs, as described in Appendix A.2.2. Surprisingly, despite the significant reduction in the training dataset size, the tokenizer in Row (iv) exhibits much stronger medical image reconstruction and classification capabilities, compared to the one in Row (iii). This highlights the importance of data quality control in training a powerful visual tokenizer[4].

$\lambda_{\text{vision}}$ **Balancing Reconstruction and Contrastive Learning.** In Rows (v) and (vi), we explores the effect of different magnitude for the visual representation alignment in the first training stage by varying $\lambda_{\text{vision}}$ in Eq. 2. By setting a light semantic constraint ($\lambda_{\text{vision}} = 0.1$), we observe an improvement across three metrics (PSNR, SSIM, and AUC) while maintaining competitive mAP, and we fix this factor in other experiments.

**Cold-Starting with Visual Representation Alignment.** In Rows (vii) and (viii), we bypass the visual representation alignment stage and train MedITok solely using the textual semantic alignment objective. While this configuration yields reasonable semantic performance, it exhibits a significant degradation in SSIM, compared with other configurations like Row (ix) of Table S1, showing the necessity of the visual pretraining stage for cold-starting MedITok by learning structural coherent representations with a light semantic constraint.

---

[4]We would like to note that this filtering was tailored to downstream tasks that mainly involve clinical images, and that other image types (*e.g.*, tables, plots, and non-clinical images) in BIOMEDICA remain highly valuable for applications that require table understanding or scientific figure interpretation.

**Freezing the Pretrained Text Encoder.** In Row (xi), we investigate the impact of unfreezing the BiomedCLIP text encoder during the second stage. Although this introduces learnable capacity into the text encoder, it disrupts the stability and alignment of the token space, leading to a trade-off: improved reconstruction metrics but severely degraded downstream classification, compared to the results in Row (xii). This suggests that freezing the pretrained textual backbone acts as an anchor, preserving the semantic information necessary for clinical interpretation.

**Visual Representation Alignment Objective.** We explore two alignment objectives for training MedITok: contrastive learning and cosine similarity (inspired by Yao et al. (2025)). Comparing Rows (xiii) and (xiv), we observe that using cosine-similarity loss yields only marginal gains in PSNR but substantially degrads downstream classification, whereas the contrastive objective produces a more discriminative token space, improving both fine-grained classification and maintaining reconstruction quality.

## D.2 ADAPTATION TO 3D MEDICAL VOLUMES

Three-dimensional data are vital in the medical domain. Our initial milestone targeted a 2D image tokenizer, considering that (1) 2D images cover more medical imaging domains (*e.g.* fundus photography, histopathology, *etc.*), (2) 2D models provide more flexibility, and (3) computational costs.

However, we note that MedITok can also be applied in 3D medical data. We compare MedITok, UniTok, and MedVAE on two 3D datasets: SLIVER07 (Heimann et al., 2009) for volume reconstruction and OrganMNIST3D (Bilic et al., 2023; Xu et al., 2019) for multi-class volume classification of 11 body organs. To adapt these 2D tokenizers to 3D volumes, we employed a slice-based strategy: processing individual slices independently and then aggregating either reconstructed slices (for reconstruction) or per-slice features (for classification). The results are summarized in Table S2.

Table S2: Additional evaluation on 3D datasets.

| Models | rFID | PSNR | SSIM | mAP | AUC |
|--------|------|------|------|------|------|
| MedVAE | 20.38 | **34.21** | **89.98** | 76.04 | 94.77 |
| UniTok | 6.89 | 31.08 | 86.16 | 83.25 | 96.15 |
| MedITok | **4.94** | 33.56 | 89.54 | **84.00** | **97.71** |

Despite not being trained explicitly on 3D radiology data, MedITok still achieves reconstruction quality comparable to MedVAE which is a *radiology-specialized* visual tokenizer, with notably lower rFID for better visual fidelity and competitive PSNR/SSIM indicating reconstruction accuracy. UniTok encodes visual semantics, yet failing to preserve critical structural details with a significant drop in PSNR and SSIM. More importantly, MedITok significantly outperforms MedVAE on 3D volume classification tasks, proving superior transferable representations in 3D settings. Visualization of 3D reconstruction results are shown in Fig. S4.

## D.3 VISUAL TURING TEST

We conducted a Visual Turing Test on the downstream medical image synthesis task, as a proxy evaluation of the quality of latent space encoded by different tokenizers. Specifically, we compare two autoregressive medical image synthesis models as in Sec. 4.4: (1) LlamaGen-MedITok, using MedITok as its visual tokenizer; and (2) LlamaGen-UniTok, using UniTok instead, a state-of-the-art unified visual tokenizer.

We randomly mixed 75 chest X-rays: 25 real, 25 synthesized by LlamaGen-MedITok, and 25 by LlamaGen-UniTok, and asked a board-certified radiologist with over 10 years' experience to score the "realness" of each image on a continuous 0–1 scale. From these scores, we computed (i) AUC for classifying real versus synthetic images and (ii) "fooling rate" or "over-confidence", the proportion of synthetic images scored higher than 0.5. As shown in Table S3, the radiologist had more difficulty distinguishing MedITok-synthesized images from real ones, indicating that MedITok enables a more clinically plausible latent space.

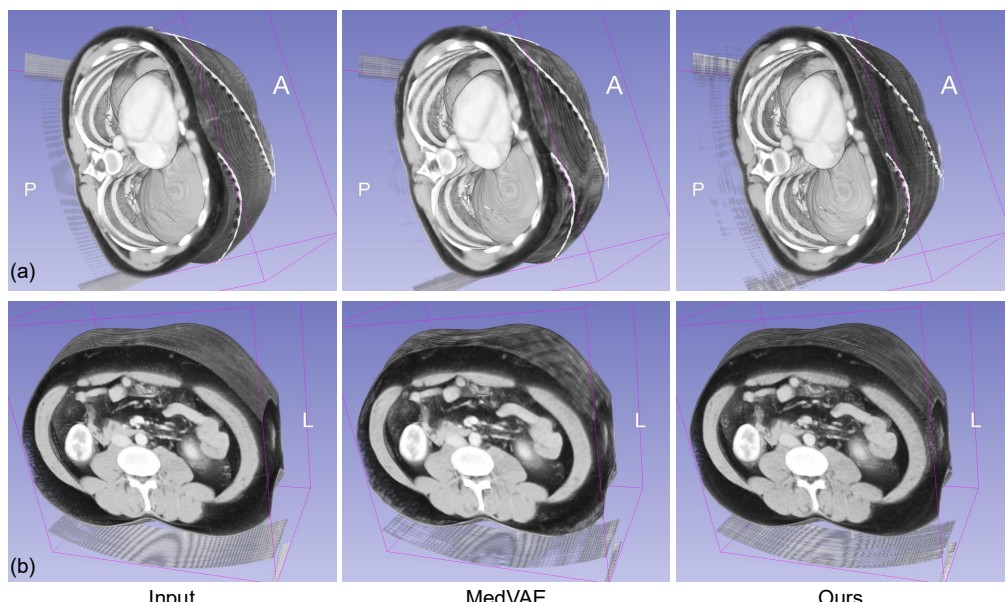

Figure S4: Visualization of 3D reconstruction results.

Table S3: Visual Turing test on downstream medical image synthesis. We report area under the ROC curve (AUC) for real vs. synthetic discrimination and the fooling rate.

| Model | AUC | Fooling rate |
|---|---|---|
| LlamaGen-UniTok | 0.602 (95% CI 0.430–0.772) | 56.0% (CI 37.1–73.3%) |
| LlamaGen-MedITok | **0.462** (95% CI 0.307–0.622) | **72.0%** (CI 52.4–85.7%) |

### D.4 DIFFERENCE BETWEEN STAGES

We compare the two stages through both performance behavior and the geometry of their learned codebooks. In Tables 3 and 4, models built upon the Stage-2 MedITok (4th row) significantly outperform those using the Stage-1 tokenizer (3rd row) in both image synthesis and interpretation, confirming that Stage 2 enhances semantic capacity without sacrificing reconstruction quality.

Empirically, Fig. S5 shows the test performance curve. In Stage 1, rFID steadily decreases while mAP remains flat or drifts slightly downward, consistent with a phase that emphasizes reconstructive accuracy over discriminative semantics. When training continues into Stage 2, mAP rises sharply, showing a strong boost in classification performance as semantic constraints are reinforced. rFID exhibits a transient increase at the first epoch in Stage 2 but then returns to a level close to the endpoint of Stage 1, indicating that reconstruction quality is largely preserved. Overall, these dynamics support the design of the two-stage schedule: Stage 1 secures a high-fidelity codebook with light semantic constraint, and Stage 2 further enhances clinical semantics in the latent vectors while retaining structural information encoding.

To understand why, we visualize the codebook vectors with $t$-SNE. As shown in Fig. S6, after Stage 2 (strong semantic alignment), the vectors spread more uniformly, pushing features to be well-distributed on the hypersphere, whereas Stage 1 (light semantic constraint) produces visibly clustered pockets.

The clustering in Stage 1 is also consistent with known VQ-VAE behavior: without additional pressures, codebooks tend to exhibit codebook collapse (Roy et al., 2018), yielding concentrated regions in latent space. The move toward a more uniform, semantically aligned latent in Stage 2 therefore explains both the stronger interpretive/synthesis performance. Notably, recent work (Yao et al., 2025) in latent diffusion reaches a congruent conclusion: aligning VAE latents to semantic-rich

features promotes generative quality by regularizing the latent geometry, with only limited impact on reconstruction.

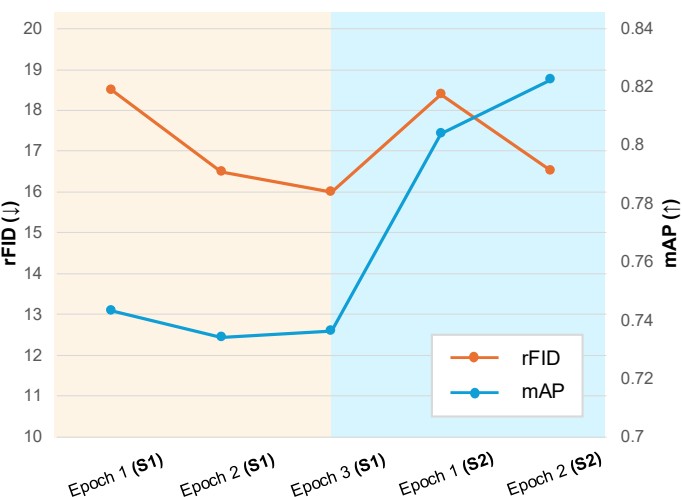

Figure S5: Training dynamics of MedITok, where rFID on the reconstruction test set and mAP on the classification test set are reported for checkpoints from three Stage-1 (S1) epochs followed by two Stage-2 (S2) epochs.

Figure S6: $t$-SNE visualization of codebook vectors in two training stages.

### D.5 FAILURE CASES

Despite the inspiring performance, MedITok may produce inferior reconstruction for histopathology images due to their rich fine-grained textures and structural complexity. As shown in the "Patho." column of Table 1, all tokenizers struggle with this modality, though MedITok still outperforms existing baselines. This represents a common challenge in histopathology tokenization that warrants future investigation. Qualitative examples for these failure cases are shown in Fig. S7.

### D.6 EFFICIENCY COMPARISON

In Table S4, we provide details on the inference GPU memory consumption (GB), and frame-per-second (FPS) throughput across different settings (*e.g.*, B8: batch size 8, R256: resolution 256). MedITok achieves comparable memory consumption and throughput to existing tokenizers while delivering state-of-the-art reconstruction quality and latent representation (Tables 1 and 2), showing both efficiency and effectiveness.

Table S4: Comparison of different models in terms of inference memory usage, and frames per second (FPS).

| Model | Memory (B16, R256) | Memory (B8, R512) | FPS (B16, R256) | FPS (B8, R512) |
|---|---|---|---|---|
| VQGAN | 3.29 | 6.31 | 136.24 | 17.76 |
| PUMIT | 0.36 | 0.56 | 4440.09 | 1691.37 |
| VAR-VQ | 4.21 | 7.97 | 171.26 | 40.95 |
| Emu3-VQ | 41.12 | OOM | 12.68 | OOM |
| VAR-VQ | 4.21 | 7.98 | 171.26 | 40.95 |
| TokenFlow | 7.91 | Not Supported | 44.15 | Not Supported |
| MedVAE | 4.61 | 8.89 | 101.56 | 24.34 |
| MedITok | 4.69 | 6.75 | 92.81 | 20.63 |

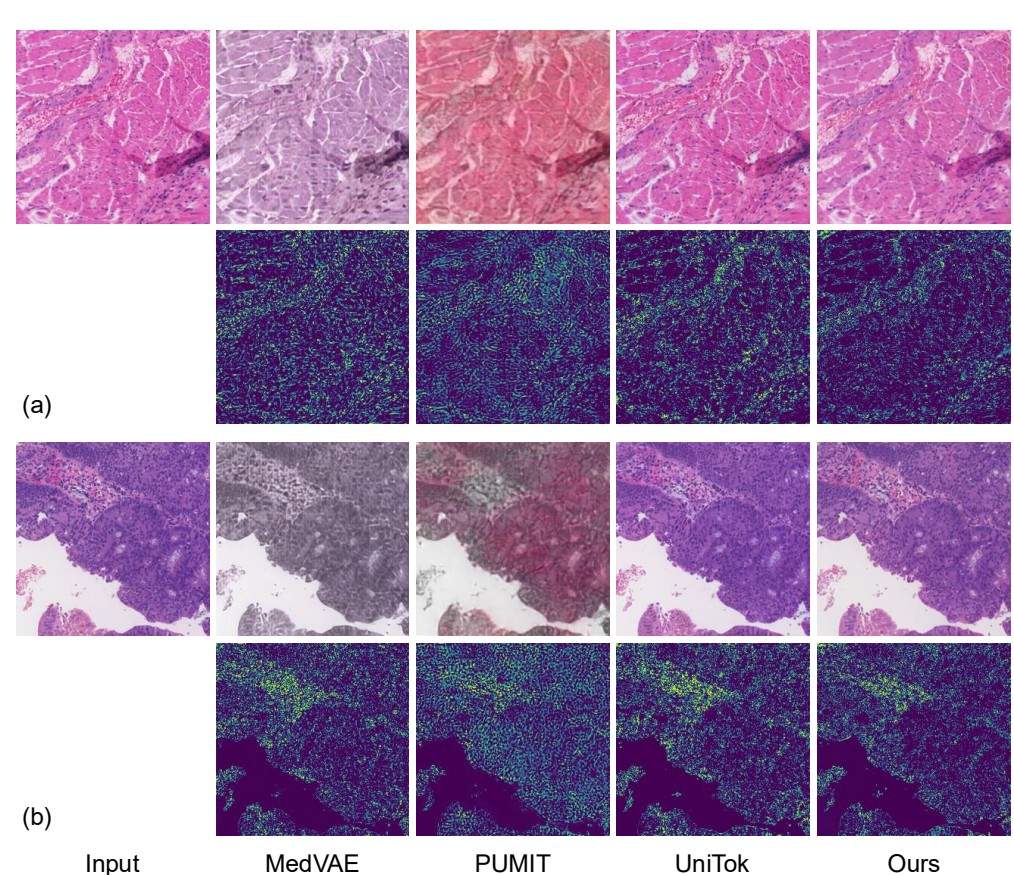

Figure S7: Two failure cases for image reconstruction. Due to the extremely rich details in histopathology images, existing visual tokenizers may still produce lower-fidelity reconstructions.

### D.7 ADDITIONAL VISUALIZATION

Fig. S8 shows more examples for qualitative evaluation of medical image reconstruction, where MedITok achieves the best visual quality with lowest errors. Fig. S9 compares the modality-conditioned synthesized images produced by different LlamaGen models. Notably, the LlamaGen model that adopts our MedITok as the visual tokenizer yields diverse and realistic medical images. Figs. S10–S12 presents the visual question answering results of LLaVA models that incoporate different visual tokenizers as their respective image encoder.

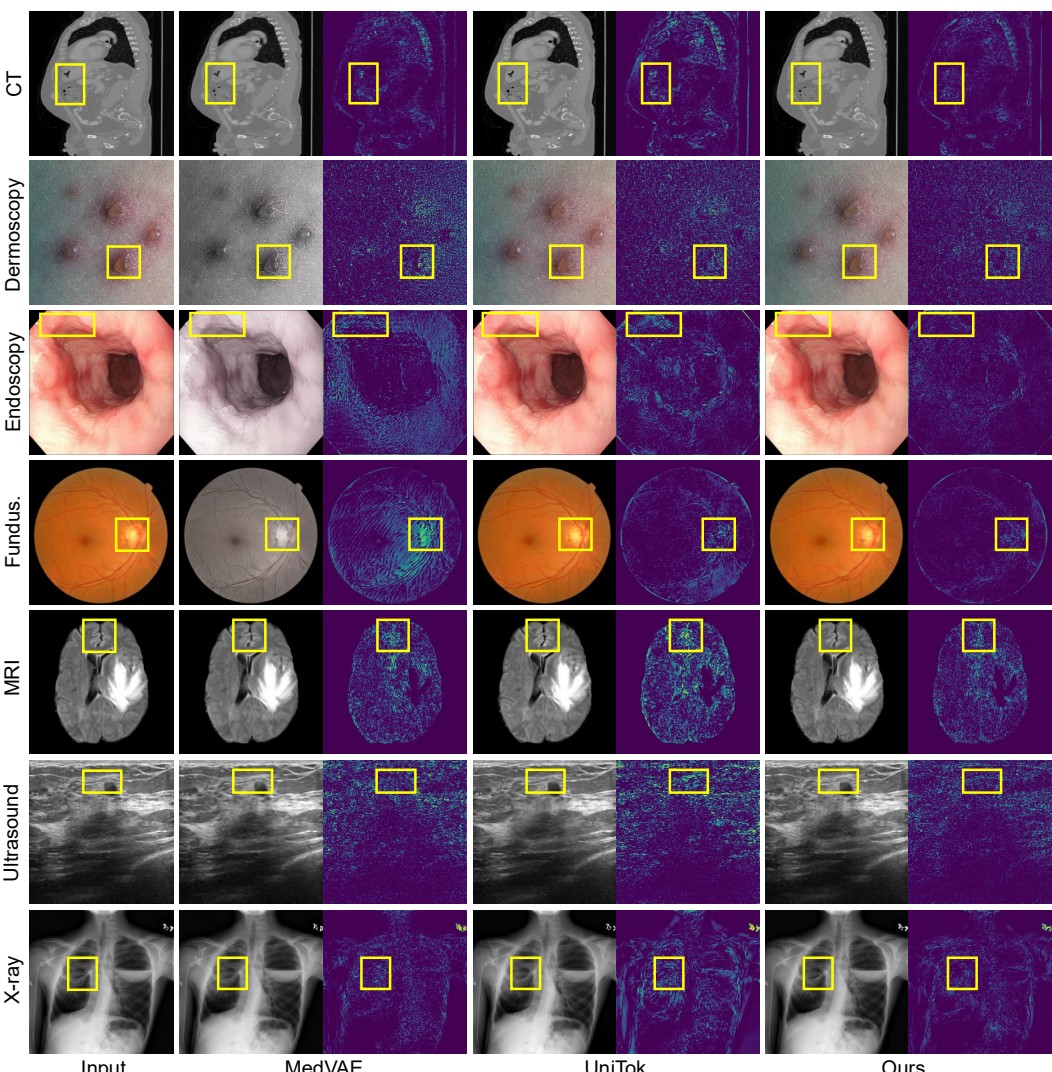

Figure S8: More reconstruction results across multiple imaging modalities. Each reconstructed image is paired with an absolute error map against the input image. Regions of interest are highlighted via yellow bounding boxes.

*Return to:* **Introduction** | **Experiments** | **Appendix Contents**

## E DISCUSSION

This section synthesizes our design choices, positioning, limitations, and societal implications. We first justify the choice of discrete codebooks (Appendix E.1). We then situate MedITok relative to representative related works, clarifying differences in objectives, training, and latent space spaces

(Appendix E.2). Next, we discuss current limitations of MedITok, and outline concrete avenues for future work (Appendix E.3). Finally, we reflect on broader impact and responsible use (Appendix E.4).

### E.1 CHOICE OF DISCRETE CODEBOOKS

Our choice of discrete tokenization is driven by the goal of building a unified latent space that can power AR models across both image synthesis and interpretation tasks in the medical domain.

To that end, discrete tokens offer the following key advantages:

- **Leveraging advances in AR modeling**: Discrete tokenization allows the medical community to directly benefit from the broader ecosystem of discrete-sequence modeling, *e.g.*, unified training objectives, any-to-any modality transfer (Zhan et al., 2024; Chen et al., 2025b), and efficient decoding and infrastructure, which are not easily transferable to continuous tokenizers.
- **Unified latent space for visual synthesis and interpretation**: Discrete tokens act as a shared representational "language" across modalities. They support AR models that can both synthesize medical images and interpret them using a single AR backbone (Lin et al., 2025). In contrast, continuous representations (*e.g.*, VAEs, CLIP) typically lack this versatility, either being hard to decode (CLIP) or poorly aligned with semantic embeddings (VAE).
- **Seamless integration with different modalities**. Discrete visual tokens are natively compatible with discrete textual tokens, enabling direct multimodal fusion in AR models without additional heads or diffusion modules. This compatibility is critical for scaling medical AR models in the style of GPT-4o, where all modalities are treated as token sequences.

### E.2 COMPARISON WITH RELATED WORKS

We situate MedITok alongside two related works: MedVAE (Varma et al., 2025) and VF-VAE (Yao et al., 2025).

MedVAE is an effective continuous variational autoencoder (VAE) designed for efficient medical image interpretation. Our primary departure from MedVAE lies in where and how semantics are bound to the latent space. Before detailing the differences, we briefly describe the training stage of interest for MedVAE and MedITok:

- MedVAE first trains a continuous VAE, then freezes the VAE encoder and decoder and learns a lightweight projector whose output is optimized so that the BiomedCLIP image embedding of the projected latent matches the embedding of the input image via an $\ell_2$ loss, *i.e.*, $\ell_2(\mathcal{E}_{\text{vision}}(f(z)), \mathcal{E}_{\text{vision}}(x))$, where $\mathcal{E}_{\text{vision}}$ denotes the pretrained BiomedCLIP vision encoder, $f$ is the projector, $x$ is the input image, and $z$ is the corresponding latent.
- MedITok utilizes $\mathcal{L}_{\text{contrastive}}(f(z), \mathcal{E}_{\text{text}}(t))$ (or $\mathcal{L}_{\text{contrastive}}(f(z), \mathcal{E}_{\text{vision}}(x))$, as in the first stage), where $\mathcal{L}_{\text{contrastive}}$ is the contrastive loss, and $t$ denotes the caption. In either stage, the encoder and decoder of MedITok are trainable.

This clearly shows the following main differences:

1. MedVAE enforces the latent $z$ to be *perceptually close* to the input image $x$ under Biomed-CLIP, which focuses more on improving the reconstruction fidelity, while MedITok aligns $z$ to the embedding space of BiomedCLIP so the MedITok *encodes more clinical semantics*.

2. MedVAE keeps the VAE encoder and decoder frozen in the second stage, which can be viewed as treating semantics as post-hoc *extraction* from a fixed latent. In contrast, MedITok *injects* semantics into a discrete token space since the encoder and decoder of the tokenizer is *both trainable*.

3. Since MedVAE focuses more on preserving structural details in radiological images, it did not utilize caption data for training and did not provide unified latent space for a wide range of downstream modalities and tasks. In contrast, by aligning latent tokens to BiomedCLIP embedding space, MedITok provides richer, fine-grained clinical semantics, which can be reflected in Table 2, where MedITok shows significantly better performance than MedVAE on image classification tasks.

Another related work is VF-VAE (Yao et al., 2025), which targets the reconstruction-generation trade-off in continuous VAE tokenizers for natural-image latent diffusion, proposing a single-stage joint reconstruction and alignment objective that aligns latents to a frozen vision foundation model to improve generative quality and training efficiency. The differences are as follows:

1. Primary task. VF-VAE focuses on improved visual generation using semantic constraint in latent diffusion, whereas our work targets unified generation and interpretation (*e.g.*, classification/VQA) across diverse medical modalities. This dual-use requirement drives our design choices.

2. Methodology design. VF-VAE employs a single-stage objective to refine the latent space for better visual generation. In contrast, we use a two-stage curriculum to reach the unified goal while exploiting abundant unlabeled medical data. Moreover, VF-VAE uses cosine similarity as the alignment objective. However, as shown in Rows (x) and (xi) of Table S1, such objective significantly degrades the medical image classification performance.

3. Architecture. VF-VAE operates in a continuous VAE/diffusion setting; MedITok produces discrete, AR-ready tokens. Architecture is not the crux here, but this helps explain downstream usage differences.

4. Community. VF-VAE contributes greatly to the field of general visual generation at designing effective VAEs. Our goal, however, is to democratize a foundation visual tokenizer for medical images to serve downstream applications, with effectiveness, scalability, and general usability for the medical image community.

### E.3 LIMITATION AND FUTURE DIRECTIONS

While MedITok demonstrates strong performance across multiple medical vision tasks, there remain important considerations and limitations that motivate future work.

*First*, our two-stage training framework effectively balances structural fidelity and semantic alignment. However, optimizing simultaneously for both properties remains non-trivial. It is interesting and valuable to explore disentangling structural and semantic objectives during training (Qu et al., 2024) or jointly optimizing the tokenizer with a downstream model that unifies visual generation and interpretation (Wang et al., 2025). We opt for the current two-stage design for its simplicity and effectiveness.

*Second*, although the current version of MedITok is designed mainly for 2D medical images across multiple imaging modalities, we have also shown that MedITok can be easily adapted to 3D medical tasks that require volume processing (Table S2). Nonetheless, MedITok could benefit from future advancement such as 3D native training or mixed training using 2D images and 3D volumes, as well as evaluation on more sophisticated tasks.

*Third*, due to resource constraints, our current experiments utilize 2.4 million image-caption pairs – modest in scale compared to billion-scale training regimes in the general domain (Ma et al., 2025b). We believe that the proposed framework is scalable and can benefit significantly from larger and more diverse image-text corpora. Future efforts may explore combining public data with institution-curated pairs.

In summary, while MedITok sets a strong foundation for unified medical visual tokenization, ongoing work is needed to address the above limitations. We envision that MedITok's flexible and expressive design can be extended to diverse downstream tasks. More broadly, we hope this work paves the way toward building scalable, general-purpose generative models that can advance medical AI and ultimately contribute to improving human health.

### E.4 BROADER IMPACT

This work presents a unified visual tokenizer tailored for medical images, offering a flexible and generalizable foundation for a wide range of medical AI applications. MedITok has the potential to accelerate the development of general-purpose medical AI systems and reduce task-specific engineering efforts. Its modular and pretrained nature also lowers the barrier for medical researchers to develop high-performance models with limited data and compute.

However, this progress also raises societal considerations. Insufficient training data may lead to biased models that underperform in underrepresented populations or clinical contexts. Additionally, the deployment of powerful downstream generative models in medicine, based on our MedITok, must be guided by strict ethical oversight to prevent misuse, misinformation, or over-reliance without clinical validation. We advocate for responsible development and interdisciplinary collaboration to ensure that such technologies benefit patients and healthcare systems.

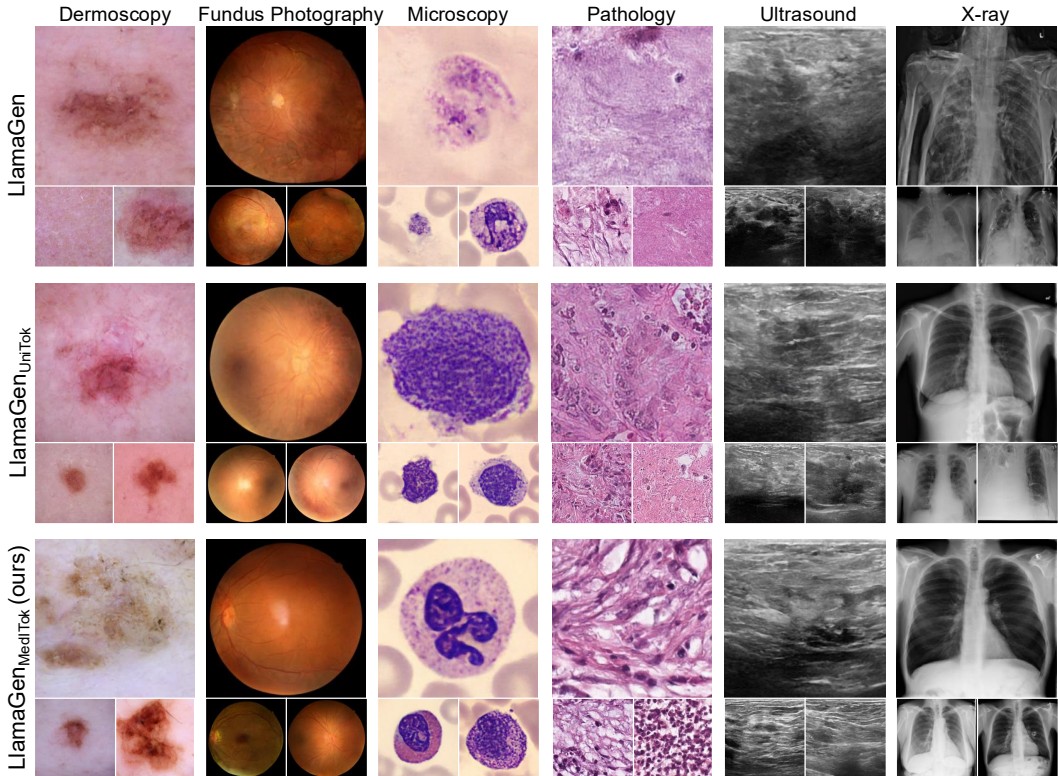

Figure S9: Qualitative comparison of medical images generated by LlamaGen models based on different visual tokenizers.

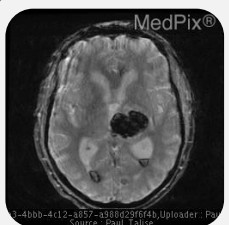

**Question**: What structures are located near the hemorrhage shown?

**Golden Answer**: Left thalamus and basal ganglia.

**LLaVA-Med**: The hemorrhage is located near the left thalamus and the left internal capsule.

**LLaVA-Med$_{UniTok}$**: The hemorrhage is located near the right cerebral peduncle, which is a part of the brainstem. This area is crucial for motor and sensory pathways, as well as autonomic functions. The presence of a hemorrhage in this region can potentially affect these critical functions.

**LLaVA-Med$_{MedITok}$ (ours)**: The hemorrhage is located near the left thalamus and the left basal ganglia. These are critical structures in the brain involved in various functions including motor control, sensory perception, and cognitive processes. The proximity of the hemorrhage to these areas could potentially impact these functions depending on the extent and severity of the bleeding.

Figure S10: Visual question answering results on an MRI image.

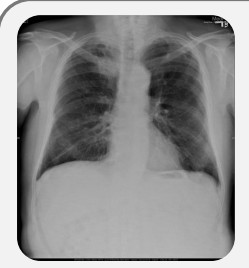

**Question**: In which lobe do you see an abnormal mass in the above images?

**Golden Answer**: Right upper lobe.

**LLaVA-Med**: The abnormal mass is located in the right upper lobe of the lung.

**LLaVA-Med_UniTok**: The abnormal mass is located in the right lower lobe of the lung.

**LLaVA-Med_MedITok (ours)**: The abnormal mass is located in the right upper lobe of the lung, as seen in the provided chest radiographs. This mass is characterized by its irregular shape and increased density compared to the surrounding lung tissue. The presence of such a mass could indicate a variety of conditions, ranging from benign tumors to malignant growths, necessitating further diagnostic evaluation such as a CT scan or biopsy to determine the exact nature and extent of the pathology.

Figure S11: Visual question answering results on a chest X-ray image.

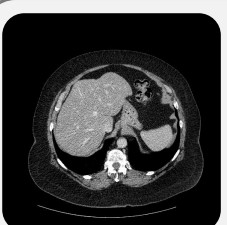

**Question**: 图片中体积最大的器官是什么?

**Golden Answer**: 肝脏.

**LLaVA-Med**: The body with the largest volume in the image is the liver.

**LLaVA-Med_UniTok**: 根据提供的CT扫描图像, 体积最大的器官是脾脏。图像显示, 脾脏的体积约为1100毫米的立方体积, 远大于其他器官的体积。脾脏的体积增大可能与某些疾病或病理状况有关, 如脾脏肿大症、脾脏炎症或脾脏肿瘤等。此外, 图像还显示肝脏、肾脏、脊柱和腹腔内的脂肪等其他器官的体积和形态。

**LLaVA-Med_MedITok (ours)**: 根据图像, 体积最大的器官是肝脏。肝脏在这张CT扫描中呈现出较为均匀的密度, 表明其内部结构相对均匀。

Figure S12: Visual question answering results on a CT image.

Table S5: Medical Image Datasets in Stage 1 (Part 1 of 4).

| Dataset Name | Count | Dataset Name | Count |
|---|---|---|---|
| Rsna-Str-Pulmonary-Embolism-Detection (Anouk Stein et al., 2020) | 5,604,627 | Bcnb-Task5 (Xu et al., 2021) | 76,559 |
| Endovis2023-Surgtoolloc (Zia et al., 2025) | 3,710,685 | Bcnb-Task1-0 (Xu et al., 2021) | 76,558 |
| Panda (Bulten et al., 2022) | 1,616,913 | Bcnb-Task6 (Xu et al., 2021) | 76,558 |
| Mela (Song et al., 2022a;b;c;d) | 1,403,843 | Msd-Liver (Antonelli et al., 2022) | 76,395 |
| Ixi (IXI) | 924,870 | Ct-Org (Rister et al., 2020) | 76,195 |
| Ribfrac2020 (Jin et al., 2020) | 810,265 | Endovis-2021-Petraw (Huaulmé et al., 2023) | 75,718 |
| Radimagenet (Mei et al., 2022) | 779,768 | Head-Neck-Pet-Ct (Vallieres et al., 2017) | 75,109 |
| Autopet (Gatidis et al., 2022) | 590,785 | Ctspine1K (Deng et al., 2021) | 72,835 |
| Brats2023-Gli (Bakas et al., 2017c) | 513,263 | Bcnb-Task1-2 (Xu et al., 2021) | 71,991 |
| Atm2022 (Zhang et al., 2023a) | 501,147 | Lndb (Pedrosa et al., 2019) | 70,292 |
| Lidc-Idri-All-Ct (Armato III et al., 2011) | 474,076 | Cptac-Hnscc (National Cancer Institute Clinical Proteomic Tumor Analysis Consortium (CPTAC), 2018) | 69,731 |
| Luna16 (Setio et al., 2017) | 431,694 | Lung-Pet-Ct-Dx (Li et al., 2020) | 66,564 |
| Brats2023-Men (LaBella et al., 2023) | 384,425 | Anti-Pd-1-Melanoma (Patnana et al., 2019) | 65,411 |
| Mimic-Cxr (Johnson et al., 2019) | 377,110 | Nsclc-Cetuximab (Movsas et al., 2016) | 64,730 |
| Qin-Headneck (Beichel et al., 2015) | 307,946 | Anode09 (Van Ginneken et al., 2010) | 63,250 |
| Biomedica (Lozano et al., 2025) | 291,155 | Opc-Radiomics | 62,726 |
| Flare22 (Ma et al., 2024) | 280,531 | Acrin-Nsclc-Fdg-Pet (Kinahan et al., 2019) | 62,701 |
| Braintumour (Bakas et al., 2018) | 263,310 | Sln-Breast (Campanella et al., 2019) | 61,968 |
| Chexpertplus (Chambon et al., 2024) | 223,460 | Bcnb-Task2 (Xu et al., 2021) | 61,828 |
| Totalsegmentator-Dataset (Wasserthal et al., 2023) | 218,477 | Msd-Lung (Antonelli et al., 2022) | 61,117 |
| Pediatric-Ct-Seg (Jordan et al., 2022; 2021) | 204,602 | Bcnb-Task1-3 (Xu et al., 2021) | 59,521 |
| Acrin6668 (Machtay et al., 2013) | 188,098 | Midrc-Ricord-1B (Tsai et al., 2021) | 59,247 |
| Covid-19-Ny-Sbu (Saltz et al., 2021) | 185,668 | Bcnb-Task1-4 (Xu et al., 2021) | 59,091 |
| Bracs (Brancati et al., 2022) | 177,712 | Learn2Reg2022-L2R-Task1-Oasis (Hering et al., 2022b) | 57,984 |
| Abdomenct1K (Ma et al., 2021a) | 172,963 | Amos2022 (Ji et al., 2022) | 56,217 |
| Bone-Marrow-Cytomorphology (Matek et al., 2021) | 171,378 | Learn2Reg22-L2R-Oasis (Hering et al., 2022b) | 52,992 |
| Ctpelvic1K (Liu et al., 2021b) | 127,315 | Cataract101 (Schoeffmann et al., 2018) | 52,676 |
| Parse22 (Luo et al., 2023a) | 122,629 | Brats2023-Ped (Kazerooni et al., 2023) | 51,769 |
| Nih-Chest-X-Rays (Wang et al., 2017a) | 112,115 | Vestibular-Schwannoma-Seg (Shapey et al., 2021) | 51,575 |
| Lits (Bilic et al., 2023) | 107,056 | Midrc-Ricord-1A (Tsai et al., 2021) | 50,913 |
| Hnscc (Grossberg et al., 2018; 2020) | 101,861 | Lc25000 (Borkowski et al., 2019) | 50,000 |
| Airogs (de Vente et al., 2024) | 101,280 | Cptac-Luad (National Cancer Institute Clinical Proteomic Tumor Analysis Consortium (CPTAC), 2018) | 48,952 |
| Head-Neck-Cetuximab (Bosch et al., 2015) | 100,356 | Ct-Covid-19-August2020 (Harmon et al., 2020) | 48,791 |
| Brats2023-Met (Moawad et al., 2023) | 93,775 | Fastpet-Ld (Green et al., 2019) | 48,097 |
| Acrin-Flt-Breast (Kinahan et al., 2017) | 91,948 | Oasis2 (Marcus et al., 2010) | 47,744 |
| Bcnb-Task4 (Xu et al., 2021) | 89,894 | Osic-Pul-Fib-Pro (Shahin et al., 2020) | 46,014 |
| Covidx-Cxr-4 (Wu et al., 2023b) | 84,802 | Anti-Pd-1-Lung (Madhavi et al., 2019) | 45,497 |
| Nlst (Team, 2011) | 79,194 | Tcga-Luad (Albertina et al., 2016) | 45,049 |
| Cad-Pe (González et al., 2020) | 78,583 | Isic2020 (Rotemberg et al., 2021) | 44,106 |
| Bcnb-Task3 (Xu et al., 2021) | 76,559 | Longitudinal-multiple-sclerosis-lesion-segmentation (Carass et al., 2017) | 41,984 |

Table S6: Medical Image Datasets in Stage 1 (Part 2 of 4).

| Dataset Name | Count | Dataset Name | Count |
|---|---|---|---|
| Covid-19-Ar (Desai et al., 2020) | 41,664 | Lysto (Jiao et al., 2024) | 19,990 |
| Glis-Rt (Shusharina & Bortfeld, 2021) | 41,143 | Cas2023 (Chen et al., 2023) | 19,200 |
| Mura (Rajpurkar et al., 2017) | 39,939 | Tcga-Ov (Holback et al., 2016) | 19,077 |
| Spie-Aapm (Armato III et al., 2015) | 39,670 | Sicapv2 (Silva-Rodríguez et al., 2020) | 18,783 |
| Tcga-Lusc (Kirk et al., 2016b) | 38,998 | Vin-Big-Data (Nguyen et al., 2020) | 17,999 |
| Atlas-2 (Liew et al., 2022) | 38,400 | Wmh (Kuijf et al., 2019) | 16,896 |
| Spie-Aapm-Lung-Ct-Challenge (Armato III et al., 2015) | 38,373 | Fizpatrick17K (Groh et al., 2021; 2022) | 16,577 |
| M2Cai16-Tool (Jin et al., 2018) | 37,314 | Chest-Image-Pneum (Zawacki et al., 2019) | 15,251 |
| Hyperkvasir (Borgli et al., 2020) | 36,329 | C-Nmc-2019 (Mourya et al., 2019) | 15,105 |
| Brats-Tcga-Gbm (Bakas et al., 2017b) | 35,770 | Covid-19-20 (Roth et al., 2022) | 15,045 |
| Lld-Mmri2023 (Lou et al., 2023) | 35,751 | Aod-14800 (Abuev, 2021) | 14,805 |
| Diabetic (Platform, 2023) | 35,059 | Aapm-Rt-Mac (Cardenas et al., 2019) | 14,080 |
| Eyepacs (Dugas et al., 2015) | 35,059 | Mindboggle (Klein et al., 2017) | 12,575 |
| Ranzcr-Clip (Seah et al., 2020) | 33,664 | Siim-Acr-Pneumothorax (Zawacki et al., 2019) | 12,053 |
| Isic2019 (Codella et al., 2018a) | 33,541 | Chest-X-Ray-Images-With-Pneumothorax-Masks (Zawacki et al., 2019) | 12,047 |
| Verse20 (Sekuboyina et al., 2021b) | 32,944 | Han-Seg (Podobnik et al., 2023) | 11,939 |
| Covidxcxr-2 (Wang et al., 2020) | 31,238 | Valdo-Task1 (Sudre et al., 2024) | 11,915 |
| Lola11 (van Ginneken, 2021) | 30,207 | Valdo-Task3 (Sudre et al., 2024) | 11,915 |
| Rsna-Pdc (Anouk Stein et al., 2018) | 29,684 | Cptac-Ucec (National Cancer Institute Clinical Proteomic Tumor Analysis Consortium (CP-TAC), 2019a) | 11,595 |
| C4Kc-Kits (Heller et al., 2019) | 28,843 | Tcga-Stad (Lucchesi & Aredes, 2016) | 11,204 |
| Word (Luo et al., 2022) | 27,154 | Ultrasound-Nerve-Segmentation (Montoya et al., 2016) | 11,143 |
| Acrin-Hnscc-Fdg-Pet-Ct (Kinahan et al., 2020) | 27,117 | Msseg08 (Styner et al., 2008) | 10,965 |
| Kits2021 (Heller et al., 2020) | 26,503 | Wsss4Luad (Han et al., 2022) | 10,091 |
| Exact09 (Lo et al., 2012) | 25,560 | Medfm-Colon-2023 (Wang et al., 2023) | 10,009 |
| Bcnb-Task1-1 (Xu et al., 2021) | 25,370 | Knee-Osteoarthritis-Dataset (Chen, 2018) | 9,766 |
| Surgvisdom (Zia et al., 2021) | 24,360 | Segthor (Lambert et al., 2020) | 9,661 |
| Brats-Tcga-Lgg (Bakas et al., 2017a) | 23,336 | Brain-Ptm (Avital et al., 2019; Nelkenbaum et al., 2020) | 9,600 |
| Tcga-Ucec (Erickson et al., 2016) | 22,946 | Msd-Colon (Antonelli et al., 2022) | 9,191 |
| Tcga-Kirc (Akin et al., 2016) | 22,644 | Covid19Ctscans (Jun et al., 2020) | 9,119 |
| Cptac-Sar (National Cancer Institute Clinical Proteomic Tumor Analysis Consortium (CP-TAC), 2019b) | 22,432 | Cholect50 (Nwoye et al., 2023) | 8,919 |
| Crossmoda2023 (Dorent et al., 2023) | 21,981 | Msd-Pancreas (Antonelli et al., 2022) | 8,666 |
| Cptac-Cm (National Cancer Institute Clinical Proteomic Tumor Analysis Consortium (CP-TAC), 2018b) | 21,867 | Fumpe (Masoudi et al., 2018) | 8,402 |
| Brats2023-Ssa (Adewole et al., 2023) | 20,910 | Lctsc (Yang et al., 2017) | 8,300 |
| Pancreas-Ct (Roth et al., 2015) | 20,709 | Ct-Vs-Pet-Ventilation-Imaging (Eslick et al., 2018) | 8,252 |
| Vessel2012 (Rudyanto et al., 2014) | 20,442 | Head-Neck-Radiomics-Hn1 (Aerts et al., 2014) | 8,161 |
| Yangxi (Liu et al., 2019) | 20,394 | Qin-Breast (Li et al., 2015) | 8,051 |
| Msseg2016 (Commowick et al., 2018) | 20,352 | Chaos-Task-4 (Kavur et al., 2021) | 7,977 |
| Oia-Odir (Peking University International Competition on Ocular Disease Intelligent Recognition (ODIR-2019), 2019) | 19,992 | Pannuke (Gamper et al., 2019; 2020) | 7,810 |

Table S7: Medical Image Datasets in Stage 1 (Part 3 of 4).

| Dataset Name | Count | Dataset Name | Count |
|---|---|---|---|
| Sppin2023 (Buser et al., 2025) | 7,616 | Pad-Ufes-20 (Pacheco et al., 2020) | 2,298 |
| Atlas2023 (Quinton et al., 2023) | 7,364 | Msd-Spleen (Antonelli et al., 2022) | 2,169 |
| Msd-Hepaticvessel (Antonelli et al., 2022) | 6,859 | Breakhis-100X (Spanhol et al., 2015) | 2,081 |
| Mmwhs (Zhuang, 2018) | 6,400 | Breakhis-200X (Spanhol et al., 2015) | 2,011 |
| Hsa-Nrl (Zhu et al., 2021) | 6,160 | Breakhis-40X (Spanhol et al., 2015) | 1,991 |
| Coronahack (Praveen Govi, 2019) | 5,933 | Breakhis-400X (Spanhol et al., 2015) | 1,820 |
| Rus-Chn (Baidu AI Studio, 2021) | 5,921 | Cptac-Pda (Consortium et al., 2018) | 1,792 |
| Dhrf (Derbi Hackathon Organizers, 2022) | 5,680 | Tiger-Wsirois-Roi-Level-Tissue-Cells (van Rijthoven et al., 2022) | 1,775 |
| Aptos2019-Blindness-Detection (apt) | 5,590 | Breast-Diagnosis (Wolberg et al., 1995) | 1,656 |
| Curious2019 (Xiao et al., 2019) | 5,376 | Cmb-Gec (Biobank, 2022a) | 1,625 |
| Cmb-Mel (Biobank, 2022b) | 5,289 | Riga-Dataset (Almazroa et al., 2018) | 1,617 |
| Clust15-2D (Luca et al., 2018) | 5,255 | Refuge2-Cls (Fang et al., 2022) | 1,600 |
| Cmmd (Cui et al., 2021) | 5,202 | Harvardglaucoma-1547 (Kim, 2018) | 1,544 |
| Tcga-Hnsc (Zuley et al., 2016) | 5,172 | Tcga-Kich (Linehan et al., 2016) | 1,484 |
| Continuous-Registration-Task3 (Baheti et al., 2021) | 5,120 | Papilledema (pap, 2020) | 1,369 |
| Messeg (Commowick et al., 2018) | 5,120 | Continuous-Registration-Task6 (Hering et al., 2022a) | 1,280 |
| Node21 (Sogancioglu et al., 2024) | 4,882 | Isbi2016-Part3 (Gutman et al., 2016) | 1,279 |
| Conic2022 (et al., 2021) | 4,870 | Isic2016-Task1 (Gutman et al., 2016) | 1,279 |
| Lag-4854 (Li et al., 2019) | 4,854 | Fusc2021 (Wang et al., 2024a) | 1,210 |
| Medfm-Chestdr-2023 (OpenMEDLab, 2023) | 4,848 | Hvsmr-2016 (Pace et al., 2015) | 1,152 |
| Stageii-Colorectal-Ct (Tong & Li, 2022) | 4,672 | Osteosarcoma-Tumor-Assessment (Leavey et al., 2019) | 1,143 |
| Naf-Prostate (Kurdziel, 2015) | 4,664 | Isic2016-Task2B-Globules (Gutman et al., 2016) | 1,142 |
| Chest-X-Ray-Pa (Asraf & Islam, 2021) | 4,574 | Isic2016-Task2B-Streaks (Gutman et al., 2016) | 1,142 |
| Lungct-Diagnosis (Grove et al., 2015) | 4,155 | Jsiec (Cen et al., 2021) | 997 |
| Covid19-Radio-Data (Chowdhury et al., 2020) | 3,886 | Isles2022 (Hernandez Petzsche et al., 2022) | 938 |
| Structseg2019-Subtask1 (Organizers, 2019) | 3,634 | Covid-19-Ct-Cxr-Det (Peng et al., 2020) | 929 |
| Structseg2019-Subtask4 (Organizers, 2019) | 3,634 | Covid-19-Ct-Cxr (Peng et al., 2020) | 918 |
| Structseg2019-Subtask2 (Organizers, 2019) | 3,413 | E-Ophta (Decenciere et al., 2013) | 905 |
| Qin-Lung-Ct (Kalpathy-Cramer et al., 2015) | 3,586 | Dao-Slocpasa (Chiu et al., 2013) | 840 |
| Structseg2019-Subtask3 (Organizers, 2019) | 3,413 | Continuous-Registration-Task5 (Klein et al., 2009) | 813 |
| Tcga-Coad (Network et al., 2012) | 3,093 | Fives (Jin et al., 2022) | 800 |
| Tcga-Prad (Abeshouse et al., 2015) | 3,007 | Segpc2021 (Gupta et al., 2023) | 773 |
| Bidr-2838 (Islam et al., 2021) | 2,838 | Paraguay-757 (Benítez et al., 2021) | 757 |
| Refuge2 (Fang et al., 2022) | 2,800 | Mudi2019 (Pizzolato et al., 2020) | 695 |
| Cptac-Ccrcc (National Cancer Institute Clinical Proteomic Tumor Analysis Consortium (CPTAC), 2018a) | 2,798 | Pulmonary-Chest-X-Ray-China (Jaeger et al., 2014a; Candemir et al., 2014b) | 662 |
| Isic2017 (Codella et al., 2018b) | 2,748 | Glaucoma-Detection (Shikamaru, 2021) | 650 |
| Verse19 (Sekuboyina et al., 2021a) | 2,650 | Beh-634 (Islam et al., 2022) | 634 |
| Palm19 (Fang et al., 2024) | 2,379 | | |

Table S8: Medical Image Datasets in Stage 1 (Part 4 of 4).

| Dataset Name | Count | Dataset Name | Count |
|---|---|---|---|
| Retina-Cataract-Dataset (yiweichen04, 2016) | 601 | Orvs (Sarhan et al., 2021) | 202 |
| Idrid (Porwal et al., 2020) | 597 | Gamma3 (Wu et al., 2023a) | 200 |
| Sz-Cxr (Stirenko et al., 2018) | 566 | Fund-179 (Yin et al., 2013) | 179 |
| Cmb-Pca (Fedorov et al., 2023) | 532 | Drac2022-Taska2 (Qian et al., 2023) | 174 |
| Crass (Hogeweg et al., 2012) | 518 | Drac2022-Taska3 (Qian et al., 2023) | 174 |
| Herlev (Jantzen et al., 2005) | 504 | Tcga-Read (Kirk et al., 2016a) | 168 |
| Papila (Kovalyk et al., 2022) | 488 | Glas (Sirinukunwattana et al., 2017) | 165 |
| Rimonedl (Batista et al., 2020) | 485 | Drac2022-Taska1 (Qian et al., 2023) | 151 |
| Fetoscopy-Placenta-Dataset (Bano et al., 2020) | 482 | Tiger-Wsirois-Roi-Level-Tissue-Bcss (Amgad et al., 2019) | 151 |
| Tcga-Blca (Kirk et al., 2016a) | 439 | Tcga-Lgg (Kirk et al., 2016a) | 145 |
| Drimdb (Prentašić et al., 2013) | 428 | Pulmonary-Chest-X-Ray-Montgomery (Jaeger et al., 2014b; Candemir et al., 2014a) | 138 |
| Toxofundus (Cardozo et al., 2023; Alam et al., 2023) | 411 | Bcss (Amgad et al., 2019) | 121 |
| Adam (Timmins et al., 2021) | 400 | Drishti-Gs-Cup (Sivaswamy et al., 2014) | 101 |
| Ph2 (Mendonça et al., 2015) | 400 | Drishti-Gs-Od (Sivaswamy et al., 2014) | 101 |
| Crown (Vos et al., 2024) | 384 | Avn (Nguyen et al., 2013) | 90 |
| Rose (Ma et al., 2021b) | 348 | Jsrt-Lung (Shiraishi et al., 2000) | 60 |
| Mias (Pisano & Yaffe, 2005) | 322 | Breast-Cancer-Cell-Seg (Gelasca et al., 2008) | 58 |
| Covid-19-Image-Dataset (Sohan, 2020) | 317 | Monuseg (Kumar et al., 2020) | 51 |
| Gamma (Wu et al., 2023a) | 300 | Hrf (Budai et al., 2013) | 45 |
| Monusac20 (Verma et al., 2021) | 283 | Drhagis (Holm et al., 2017) | 40 |
| Rod (Grace Maria Binu, 2023) | 281 | Drive (Staal et al., 2004) | 40 |
| Jsrt (Shiraishi et al., 2000) | 247 | Rite (Hu et al., 2013) | 40 |
| Jsrt-Gender-Cls (Shiraishi et al., 2000) | 247 | Hrf-Quality-Cls (Budai et al., 2013) | 36 |
| Tcga-Sarc (Kirk et al., 2016a) | 241 | Retinacheck (Dashtbozorg et al., 2016) | 30 |
| Crag (Graham et al., 2019a) | 213 | Olives-Fundus-Photography (Prabhushankar et al., 2022) | 14 |
| Panda-Radboud (Nir et al., 2018a) | 206 | Occmcpv (Chen et al., 2024a) | 8 |

Table S9: Medical Image Datasets in Stage 2.

| Dataset Name | Count | Dataset Name | Count |
|---|---|---|---|
| Biomedica (Lozano et al., 2025) | 1,216,529 | Mimic-Cxr (Johnson et al., 2019) | 107,684 |
| Gmai-Vl-5.5M (Li et al., 2024) | 671,824 | Rocov2 (Rückert et al., 2024) | 59,212 |
| Medicat (Subramanian et al., 2020) | 204,772 | Pmc-Oa (Lin et al., 2023) | 36,386 |
| Llava-Med-Instruct-Fig-Captions (Li et al., 2023) | 122,843 | Mm-Retinal (Wu et al., 2024) | 3,577 |

Table S10: Medical Image Datasets for Image Reconstruction Evaluation.

| Dataset Name | Count | Dataset Name | Count |
|---|---|---|---|
| Ivygap-Radiomics (Pati et al., 2020) | 8,456 | Monkeypox (Ali et al., 2022) | 802 |
| Chestx-Det (Lian et al., 2021) | 3,578 | Breast-Ultrasound-Images-Dataset (Al-Dhabyani et al., 2020) | 647 |
| Aapm-lowdose-ct (McCollough et al., 2017) | 3,413 | Ddti (Pedraza et al., 2015) | 637 |
| Btcv-Cervix (Landman et al., 2015) | 3,039 | Hie2023 (Bao et al., 2025) | 554 |
| Surgt (Cartucho et al., 2024) | 2,933 | Digestpath19-Cls (Da et al., 2022) | 455 |
| Silver07 (Heimann et al., 2009) | 2,291 | EndoCV2020-EDD (Ali et al., 2020) | 386 |
| Derm7Pt (Kawahara et al., 2018) | 2,013 | Mednode (Giotis et al., 2015) | 170 |
| Messidor (Decencière et al., 2014) | 1,748 | Gleason (Nir et al., 2018b) | 103 |
| Rsna-Bone-Age (Halabi et al., 2019) | 1,596 | Consep (Graham et al., 2019b) | 41 |
| Hmc-Qu (Kiranyaz et al., 2020) | 1,269 | Chase (Fraz et al., 2012) | 28 |
| Covidgr (Tabik et al., 2020) | 852 | Stare (Hoover et al., 2000) | 20 |

Table S11: Downstream Medical Vision Tasks Datasets. "CLS": classification. "M2I": modality-to-image synthesis. "VQA": visual question answering.

| Dataset | Train | Test | Modality | Task Type | Classes |
|---|---|---|---|---|---|
| PneumoniaMNIST (Kermany et al., 2018) | 4,708 | 1,148 | X-ray | CLS | 2 |
| PathMNIST (Kather et al., 2019) | 89,996 | 500 | pathology | CLS; M2I | 9 |
| ChestMNIST (Wang et al., 2017b) | 78,468 | 500 | X-ray | M2I | 14 |
| BloodMNIST (Acevedo et al., 2020) | 11,959 | 500 | microscopy | M2I | 8 |
| DermaMNIST (Tschandl et al., 2018; Codella et al., 2019) | 7,007 | 500 | dermoscopy | CLS; M2I | 7 |
| RetinaMNIST (Liu et al., 2022) | 1,080 | 500 | fundus photography | CLS; M2I | 5 |
| BreastMNIST (Al-Dhabyani et al., 2020) | 546 | 234 | ultrasound | CLS; M2I | 2 |
| Pubmed-Vision-Caption (Chen et al., 2024c) | 555,103 | 0 | Unknown | VQA | – |
| Pubmed-Vision-VQA (Chen et al., 2024c) | 100,000 | 0 | Unknown | VQA | – |
| VQARAD-Test (Lau et al., 2018) | 0 | 451 | Unknown | VQA | – |
| Slake-Test (Liu et al., 2021a) | 0 | 2,094 | Unknown | VQA | – |
| Slake-Val (Liu et al., 2021a) | 0 | 2,099 | Unknown | VQA | – |

