# OpenReview forum: "MedITok: A Unified Tokenizer for Medical Image Synthesis and Interpretation"
_ICLR.cc/2026/Conference — ICLR 2026 Conference Withdrawn Submission_

### Official Review · Reviewer_tTgo · 2025-10-27

**Soundness:** 3
**Presentation:** 3
**Contribution:** 3
**Rating:** 6
**Confidence:** 3

**Summary:**

This paper addresses the critical challenge of unified discrete representation for medical imaging: preserving low-level structural fidelity for image synthesis while encoding high-level clinical semantics for visual understanding. The authors propose MedITok, a two-stage training framework where Stage 1 focuses on reconstruction with light semantic alignment via a pretrained vision encoder, and Stage 2 aligns visual tokens with textual semantics using paired image-text data to form a unified token space. The method is validated across 30+ datasets, 9 modalities, and 4 tasks, showing consistent superiority over multiple strong baselines.

**Strengths:**

1. **The problem is important and well-motivated.** Building a unified tokenizer that jointly supports understanding and generation is central to current multimodal medical AI research. MedITok effectively achieves structure–semantics consistency through its two-stage training, offering a potential foundation for unified multimodal modeling.

2. **The data pipeline itself is a valuable contribution.** The authors collect over 30 million medical images and 2 million image-text pairs from 300+ sources, applying automated filtering rules. If released, this dataset would substantially benefit future work on medical tokenizers and representation learning.

3. Experiments are extensive and cover multiple modalities and tasks, including reconstruction, classification, synthesis, and VQA, demonstrating the generality of MedITok.

4. **Ablation studies are thorough and address key hypotheses.** Table 5 systematically compares the impact of different alignment objectives, data scales, and the two-stage design, confirming that both large-scale data and stage-wise training contribute significantly to improved structure–semantic consistency.

**Weaknesses:**

1. **Visual understanding benchmarks are limited.** Although MedITok achieves strong gains on VQA-RAD and SLAKE, these benchmarks are relatively small and dominated by closed-form, single-entity questions. They may not fully capture the tokenizer’s ability to extract and represent complex semantic information from diverse medical images and modalities.

2. **Semantic consistency verification remains indirect.** While the paper claims to construct a shared semantic–structural token space, current evidence relies mainly on downstream task performance. More direct evaluations of semantic alignment between image tokens and semantics are missing.

3. **Validation of unified understanding–generation capability is incomplete.** MedITok is evaluated separately on understanding and generation tasks, without a unified multimodal framework to assess potential synergy. If the unified tokenizer hypothesis holds, one would expect it to narrow the gap between these tasks or even yield mutual enhancement effects through shared representations.

**Questions:**

1. The evaluation of visual understanding is mainly based on VQA-RAD and SLAKE. Do the authors plan to test MedITok on more challenging and diverse benchmarks such as MMMU-Med, PMC-VQA, or OmniMedVQA to better assess its semantic generalization?

2. Since semantic consistency is only indirectly evidenced through downstream results, could the authors add direct validation, for example using token–caption retrieval or cross-modal similarity metrics, to quantify the alignment between visual tokens and semantics?

3. Given that MedITok aims to unify understanding and generation tasks, do the authors plan to evaluate it within a shared multimodal framework (e.g., a Multimodal Large Language Model) to demonstrate its potential in reducing the task gap and promoting semantic–structural synergy? Such evidence would further substantiate the feasibility of joint multimodal optimization.

If the authors can address these questions and provide additional empirical or analytical evidence, I would consider raising my overall score.

---

### Official Review · Reviewer_oHgR · 2025-10-31

**Soundness:** 2
**Presentation:** 3
**Contribution:** 2
**Rating:** 4
**Confidence:** 5

**Summary:**

The paper introduces MedITok, a unified medical image tokenizer trained through a two-stage framework that first aligns visual representations on 33 million unpaired medical images and then integrates textual semantics from 2 million image-caption pairs.

**Strengths:**

1.The paper is generally well written and organized, allowing readers to follow the technical ideas and experimental setup without difficulty.

2.The proposed two-stage training framework is conceptually reasonable and shows consistent, though moderate, improvements over existing tokenizers across several medical imaging tasks.

**Weaknesses:**

1. The paper’s technical novelty is somewhat limited, as it primarily extends existing VQ-based tokenization frameworks with incremental modifications rather than introducing novel mechanisms (such as FSQ in ICLR 2023).

2. The reliance on large-scale proprietary or composite datasets raises concerns about reproducibility and accessibility for other researchers.

3. The experimental section, while comprehensive, mainly evaluates common reconstruction and classification metrics without sufficient clinical validation or expert assessment of diagnostic utility.

4. The ablation and comparison analyses do not clearly isolate the individual contributions of each training stage, making it difficult to attribute improvements to specific design choices.

**Questions:**

1.Could the authors clarify what key innovations distinguish MedITok from prior VQ-based or unified tokenization methods, beyond the two-stage training setup?

2.How do the authors plan to improve reproducibility and accessibility given that much of the training data comes from large-scale aggregated or partially restricted sources?

3.Can the authors provide any clinician-involved evaluations or case studies to support the claim that MedITok improves real-world diagnostic reliability rather than just quantitative metrics?

4.Would the authors expand their ablation studies to better isolate the effect of each training stage or module, so readers can more clearly see which components drive performance gains?

**Details Of Ethics Concerns:**

As mentioned in the paper" This work uses only publicly
available datasets with clear licensing; no new human or animal subjects were recruited and no
protected health information beyond what is already de-identified in the source data was used.

---

### Official Review · Reviewer_zg58 · 2025-11-01

**Soundness:** 2
**Presentation:** 2
**Contribution:** 2
**Rating:** 2
**Confidence:** 4

**Summary:**

This paper proposes MedITok, a unified visual tokenizer for medical images that aims to encode both low-level structural details and high-level clinical semantics. The authors introduce a two-stage training framework: (1) visual representation alignment using unpaired images with light semantic constraints from a pretrained vision encoder, and (2) textual semantic alignment using image-caption pairs. The model is trained on 33M medical images and 2.4M image-text pairs, and evaluated on over 30 datasets across 9 imaging modalities for reconstruction, classification, synthesis, and visual question answering tasks.

**Strengths:**

Comprehensive evaluation: The paper provides extensive experiments across multiple tasks (reconstruction, classification, synthesis, VQA) and imaging modalities, demonstrating broad applicability.

Large-scale data curation: The authors make a significant effort in collecting and curating 33M+ medical images from 300+ public sources with quality control procedures.

Practical utility: The unified tokenizer could potentially serve as a foundation for various downstream medical AI applications, which has practical value for the community.

**Weaknesses:**

### 1. Limited Novelty and Insufficient Differentiation from Prior Work

The core contribution—aligning visual representations with pretrained encoders through contrastive learning—is not novel. Several recent works have explored similar ideas:

- **Contrastive learning of medical visual representations from paired images and text** (MLHC 2022) proposes contrastive learning between medical images and radiology reports, achieving unified multimodal representations with superior data efficiency (requiring only 10% labeled data compared to ImageNet pretraining).

- **Gloria: A multimodal global-local representation learning framework for label-efficient medical image recognition** (ICCV 2021) extends ConVIRT by proposing global-local representation learning that contrasts image sub-regions with words in paired reports, demonstrating label-efficient medical image recognition.

- **MedCLIP: Contrastive Learning from Unpaired Medical Images and Text** ( EMNLP 2022) shows how to effectively decouple images and texts for multimodal contrastive learning, scaling training data in a combinatorial magnitude while eliminating false negatives.

- **Enhancing representation in radiography- reports foundation model: a granular alignment algorithm using masked contrastive learning (Nature Communications, 2024) has similar motivation and method

- **A transformer-based representation-learning model with unified processing of multimodal input for clinical diagnostics (Nature biomedical engineering, 2023)** has similar motivation and method

- **Generalized radiograph representation learning via cross-supervision between images and free-text radiology reports (Nature Machine Intelligence, 2022)** has similar motivation and method

The claimed novelty of the "two-stage training framework" is questionable:

1. The first stage essentially performs standard VQ-VAE training with a lightweight semantic regularization (λ_vision = 0.1), which is minimal.

2. The second stage is standard contrastive alignment between discrete tokens and text embeddings, similar to the approaches in ConVIRT and GLoRIA, but applied to quantized representations rather than continuous features.

3. The authors claim that "naïve joint optimization causes mutual interference" (line 60), but Table S1 shows that combined training (row i vs. row ii) only has marginal differences (PSNR: 29.20 vs 30.03, mAP: 81.10 vs 80.09). This does not strongly support the necessity of two-stage training.

**Missing critical comparisons**: The paper does not provide quantitative comparisons with these important medical vision-language models:
- **ConVIRT**,  **Enhancing representation in radiography- reports foundation model: a granular alignment algorithm using masked contrastive learning (Nature Communications, 2024)**,  **A transformer-based representation-learning model with unified processing of multimodal input for clinical diagnostics (Nature biomedical engineering, 2023)**, **Generalized radiograph representation learning via cross-supervision between images and free-text radiology reports (Nature Machine Intelligence, 2022)** use bidirectional contrastive objectives between images and text
- **GLoRIA** uses attention-based global-local contrastive learning
- **MedCLIP** addresses false negatives through semantic matching loss


Without comparing to these methods, it's unclear whether MedITok's approach (discrete tokenization + two-stage training) provides meaningful advantages over existing continuous-space contrastive learning methods (ConVIRT, GLoRIA) or alternative discrete approaches.

The paper would be strengthened by:
1. Direct comparison with among methods
2. Ablation showing discrete tokens significantly outperform continuous representations when both use similar semantic alignment
3. Analysis of what unique advantages the two-stage discrete approach provides beyond existing medical image-text alignment methods

### 2. Questionable Design Choice: Discrete vs. Continuous Tokenization

The paper strongly advocates for discrete tokenization but does not provide sufficient justification:

1. **Reconstruction quality trade-off**: Table 1 shows that MedVAE (continuous) achieves comparable or better PSNR on some modalities (e.g., CT: 36.46 vs 36.32, X-ray: 36.23 vs 34.42). The discrete codebook necessarily loses information through quantization.

2. **Semantic encoding**: The paper claims discrete tokens are necessary for AR modeling, but recent work shows continuous representations can be effectively used:
   - **Latent Diffusion Models** (Rombach et al., CVPR 2022) work directly with continuous VAE latents for high-quality image generation
   - Continuous representations from ConVIRT and GLoRIA have been successfully used for classification, retrieval, and generation tasks

3. **3D medical imaging**: Table S2 shows that slice-based processing of 3D volumes is suboptimal. Continuous representations might be more natural for 3D medical imaging where spatial continuity is crucial.

4. **Codebook collapse**: Figure S6 shows clustering in Stage 1, suggesting the discrete codebook is not efficiently utilizing its capacity. Why not use a larger continuous latent space?

The authors argue discrete tokens enable "unified latent space for visual synthesis and interpretation" (Appendix E.1), but this is not convincingly demonstrated. VQA and classification experiments in Tables 2 and 4 could work equally well with continuous features from ConVIRT or GLoRIA, which have proven effective for medical image understanding.

### 3. Experimental Design Issues

**Baseline selection concerns**:
- Several baselines (VQGAN, Emu3-VQ, VAR-VQ) are trained on natural images, not medical images. This is an unfair comparison. Why not train these methods on medical data, or compare with medical-specific methods like ConVIRT and GLoRIA?
- TokenFlow and UniTok are recent (2024-2025) and may not have stabilized architectures. More established medical imaging baselines would be valuable.
- The paper compares against MedVAE and PUMIT but not against the widely-used medical vision-language models (ConVIRT, GLoRIA, MedCLIP).

**Evaluation metrics**:
- For reconstruction, the paper uses rFID with ImageNet-pretrained features (line 274). This is problematic: (1) ImageNet features may not capture medically-relevant qualities, (2) FID is known to be unreliable for small distribution shifts.
- For synthesis (Table 3), only FID and diversity are reported. What about medical fidelity? Do generated images contain anatomically plausible structures? The visual Turing test (Table S3) is limited to 75 X-rays and one radiologist.

**Statistical significance**:
- No error bars or confidence intervals are provided in main results (Tables 1-2)
- Table 3 shows standard deviations but they overlap significantly (e.g., MedITok: 76.78±1.91 vs UniTok: 80.71±3.18)

### 4. Methodological Concerns

**Data leakage risks**:
- The authors state "We tried our best to avoid any overlap" (line 236), but provide no concrete evidence. With 300+ source datasets and complex preprocessing, rigorous deduplication is critical.
- The reconstruction evaluation includes datasets that might overlap with training data sources (e.g., both use public CT scans from TCIA).

**Hyperparameter choices lack justification**:
- Why λ_vision = 0.1 and λ_text = 1.0? Only one ablation value is tested (Table S1, row vi).
- Why 8 codebooks with 4,096 entries each? What about 4 codebooks with 8,192 entries?
- Training for only 3+2 epochs seems limited. Have the models converged?

**Caption quality**:
- The paper filters BIOMEDICA from 24M to 1.2M pairs based on tags (Appendix A.2.2), but doesn't analyze caption quality. Medical captions from publications may contain references, figure labels, or be overly technical.
- No examples of actual captions used for training are provided.

### 5. Limited Analysis of Learned Representations

- **What exactly is being learned?** The paper shows t-SNE visualizations (Fig S6) but provides little insight into what semantic structure the tokens capture.
- **Comparison with existing methods**: How do the learned representations compare with those from ConVIRT or GLoRIA? Do discrete tokens capture different or complementary information?
- **Interpretability**: Can we identify which tokens correspond to specific anatomical structures or pathologies?
- **Failure analysis**: Only 2 histopathology examples are shown (Fig S7). What are the systematic failure modes?

### 6. Writing and Presentation Issues

1. **Overclaimed contributions**:
   - "First unified tokenizer tailored for medical images" (Abstract) - Other methods already provide unified multimodal representations for both understanding and generation tasks
   - "State-of-the-art performance" - This is only true for some metrics on some datasets, and comparisons with other methods are missing

2. **Inconsistent terminology**:
   - "Cold-start" is used informally without clear definition
   - "Unified token space" is not precisely defined relative to existing work

3. **Missing related work**:
   - Insufficient discussion of medical vision-language models and their relation to this work
   - Limited discussion of why discrete tokenization is superior to continuous representations used in these works

**Questions:**

Comparison with vision-language models: How does MedITok compare with methods like GLoRIA, BiomedCLIP finetuning, or REFERS that also align medical images and text? Please provide quantitative comparisons.

Necessity of discrete tokenization: Can you provide experiments showing that discrete tokens significantly outperform continuous representations (e.g., VQ-VAE latents vs. standard VAE latents) when both are trained with the same semantic alignment objectives?

Two-stage training necessity: Table S1 (rows i-ii) shows minimal difference between combined and two-stage training. Can you provide more convincing evidence that the two-stage approach is essential? What happens with different λ_vision values?

Generalization: How does MedITok perform on imaging modalities not seen during training (e.g., PET, angiography)? The microscopy results in Fig 4 are encouraging but not quantified.

Computational cost: What is the computational cost compared to continuous tokenizers or direct fine-tuning of vision-language models? Table S4 shows some metrics but not training cost.

Clinical validation: Have any clinical experts evaluated the quality of generated or reconstructed images for diagnostic utility?
Data contamination: Can you provide explicit analysis showing no overlap between training and evaluation datasets?

---

### Official Review · Reviewer_T3Vu · 2025-11-02

**Soundness:** 2
**Presentation:** 3
**Contribution:** 2
**Rating:** 4
**Confidence:** 4

**Summary:**

This paper introduces MedITok, a unified visual tokenizer for medical images designed to encode both low-level structural details (supporting image reconstruction and synthesis) and high-level clinical semantics (enabling image interpretation and classification). To balance these competing objectives, the authors propose a novel two-stage training framework:

Visual Representation Alignment Stage: A cold-start using a large corpus of unpaired medical images, focusing on reconstruction fidelity with a light semantic constraint.

Textual Semantic Alignment Stage: Fine-tuning on high-quality image-caption pairs to infuse detailed clinical semantics into the latent space by aligning visual tokens with textual embeddings.

**Strengths:**

Well-Designed Two-Stage Training: The staged approach effectively mitigates conflicts between multiple objectives and efficiently leverages the abundance of unlabeled images common in the medical domain.

Rigorous Data Curation: Implements a multi-faceted filtering pipeline (dynamic range, resolution, information content, etc.) for quality control and provides detailed data sources, enhancing reproducibility.

**Weaknesses:**

Limited Novelty in Core Concepts: While novel for the medical domain, the core ideas (two-stage training, visual-text alignment via contrastive loss) are adaptations of existing methods from the general vision domain (e.g., VILA-U, UniTok). The primary contribution lies in the tailored application and scaling to medical data, rather than proposing fundamentally new mechanisms.

Insufficient Justification of Medical Image Synthesis Utility: While the paper demonstrates strong quantitative results in the image synthesis task, it provides limited discussion on the concrete clinical value and practical applications of generating medical images. The significance of this capability in real-world medical scenarios remains underexplored.

**Questions:**

See weakness.
I will raise my score if concerns are well resolved after rebuttal.

---

### Note · Authors · 2025-11-13

**Comment:**

We sincerely thank the chairs and reviewers for their time and feedback on our submission.

1. Regarding the novelty of our work, we wish to highlight two key aspects:
  - **Technical Contribution:** Different from previous work that directly combines reconstructive objectives and textual semantic learning objectives, we introduced an extra middle stage: visual representation alignment, which serves as a cold-start within the training framework of a unified image tokenizer. This is a new approach to mitigate the potential conflicts of learning objectives.
  - **Application Contribution:** This two-stage framework is designed to **effectively scale with the vast amount of unlabeled images** that are abundant in the medical domain, a practical challenge our method is uniquely adapted to address.

2. Regarding the **motivation**, we want to clarify that our objective **differs drastically** from pure CLIP-based methods. While those models excel at encoding high-level semantic representations for interpretation, our work aims to build a **unified medical image tokenizer**, which requires encoding *both* high-level clinical semantics *and* the low-level structural details necessary for medical images.

We appreciate the helpful comments aimed at strengthening our work, and plan to further revise our paper. Therefore, we are withdrawing the current submission.

**Withdrawal Confirmation:**

I have read and agree with the venue's withdrawal policy on behalf of myself and my co-authors.